# A unique *Toxoplasma gondii* haplotype accompanied the global expansion of cats

Lokman Galal [1] ✉, Frédéric Ariey[2], Meriadeg Ar Gouilh [3,4,5], Marie-Laure Dardé [1,6], Azra Hamidović[1], Franck Letourneur [7], Franck Prugnolle [8] & Aurélien Mercier [1,6] ✉

*Toxoplasma gondii* is a cyst-forming apicomplexan parasite of virtually all warm-blooded species, with all true cats (Felidae) as definitive hosts. It is the etiologic agent of toxoplasmosis, a disease causing substantial public health burden worldwide. Few intercontinental clonal lineages represent the large majority of isolates worldwide. Little is known about the evolutionary forces driving the success of these lineages, the timing and the mechanisms of their global dispersal. In this study, we analyse a set of 156 genomes and we provide estimates of *T. gondii* mutation rate and generation time. We elucidate how the evolution of *T. gondii* populations is intimately linked to the major events that have punctuated the recent history of cats. We show that a unique haplotype, whose length represents only 0.16% of the whole *T. gondii* genome, is common to all intercontinental lineages and hybrid populations derived from these lineages. This haplotype has accompanied wildcats (*Felis silvestris*) during their emergence from the wild to domestic settlements, their dispersal in the Old World, and their expansion in the last five centuries to the Americas. The selection of this haplotype is most parsimoniously explained by its role in sexual reproduction of *T. gondii* in domestic cats.

*Toxoplasma gondii* is a zoonotic protozoan that has spread globally. This apicomplexan parasite infects all warm-blooded species, including humans, and its wide range of host species suggests multiple routes for short and long-distance parasite migrations[1]. *T. gondii* is found in ~30% of the human population and is the etiologic agent of toxoplasmosis, a disease causing a substantial public health burden worldwide[2]. Infection with *T. gondii* has been long considered benign, or even asymptomatic, except for certain risk groups like the developing foetus in case of congenital infection with 200,000 new cases each year[3], and immunocompromised patients for whom toxoplasmosis can have severe health consequences either during primo-infection or reactivation. However, certain *T. gondii* populations have been associated with severe toxoplasmosis in immunocompetent individuals[4–6]. More importantly, an increasing number of epidemiological studies suggest that chronic infection with *T. gondii* is associated with a wide variety of neuropsychiatric disorders, substantially raising the public health importance of this global and highly prevalent parasite[7]. Given gaps in both the current preventive (no vaccine available for humans) and therapeutic strategies[8,9], active research to discover new ways to target this clinically important protozoan is still needed.

[1]Inserm U1094, IRD U270, Univ. Limoges, CHU Limoges, EpiMaCT—Epidemiology of chronic diseases in tropical zone, Institute of Epidemiology and Tropical Neurology, OmegaHealth, Limoges, France. [2]Université de Paris, Institut Cochin, Inserm U1016, Service de Parasitologie Hôpital Cochin, 75014 Paris, France. [3]DYNAMICURE U1311 INSERM, Université de Caen Normandie, UNICAEN, UNIROUEN, 14000 Caen, France. [4]Laboratoire de Virologie, Centre Hospitalo-Universitaire, Avenue Georges Clémenceau, 14000 Caen, France. [5]Institut de Recherche pour le Développement (IRD), Maladies Infectieuses et vecteurs: Ecologie, Génétique, Evolution et Contrôle (MIVEGEC) (Université de Montpellier—IRD 224—CNRS 5290), Montpellier, France. [6]Centre National de Référence (CNR) Toxoplasmose/Toxoplasma Biological Center (BRC), Centre Hospitalier-Universitaire Dupuytren, Limoges, France. [7]Plate-Forme Séquençage et Génomique, Institut Cochin, Inserm U1016, Université de Paris, 22 rue Méchain, 75014 Paris, France. [8]IRL REHABS, International Research Laboratory REHABS, CNRS-NMU-UCBL, Nelson Mandela University George Campus, George 6531, South Africa. ✉e-mail: lokmanmagalal@gmail.com; aurelien.mercier@unilim.fr

*T. gondii* hosts get infected after ingestion of oocysts shed into the environment by contaminated faeces of felids. Another source of infection for humans and other meat-consuming species is the ingestion of raw or undercooked meat from animals harbouring infective tissue cysts. In the domestic environment, cats and rodents are considered the most significant reservoirs for human infection, since life cycle completion relies mainly on transmission between these two categories of animal hosts, the rodents being the main prey of cats[10,11]. Sexual recombination is possible when two different strains are found simultaneously in the cat's gut. For this to occur, a cat has to ingest within a few hours two prey infected with different strains. There is, therefore, a time barrier for recombination to occur, or alternatively, the cat has to ingest a single prey infected with two different strains (mixed infection), a rare event in nature given that intermediate hosts develop immunity to new infections following their first infection.

From a genetic point of view, the population structure of *T. gondii* is characterized by contrasting patterns of strain diversity, mainly varying according to geographical origin and ecotype. To date, most studies of diversity have relied on the analysis of microsatellite markers (MS), or Restriction Fragment Length Polymorphism (RFLP) (refer to Supplementary Data 1 for correspondence MS and RFLP designations of lineages or genotypes). In the Old World (Africa, Asia and Europe), most *T. gondii* isolates from humans, domestic animals and wild fauna were found to belong to a few intercontinental clonal lineages: type I, type II, type III, Africa 1 (also designated as BrI) and Africa 4[12,13]. Few other clonal lineages have been described in certain countries, such as Chinese 1 in China[14] and Africa 3 in Gabon[15], and strains not belonging to these major lineages have been rarely isolated[16]. This genetic evidence argues that sexual recombination between different strains is very rare in these regions. In the New World (North and South America), the most common Old World clonal lineages (type I, II, III and Africa 1) are also found beside a diversity of local clonal lineages and non-clonal populations specific to South or North America[12,17]. In contrast to the pattern observed in the Old World, the genotypic composition of strains from wildlife differs importantly from strains commonly isolated in the domestic environment[17,18]. These *T. gondii* populations—isolated from wild animals or from humans in contact with wildlife and genetically distinct from *T. gondii* populations found in the domestic environment—are associated with environments where the presence of wild felids is well-established[1,4,18,19]. This observation supports the notion that specific co-adaptations have occurred between *T. gondii* strains and different feline species[20–22].

To date, population genetic studies have only partially deciphered the phylogenetic relationship between strains from different geographical areas. In particular, the phylogenetic positioning of the most common intercontinental clonal lineages relative to other *T. gondii* lineages and populations remains unclear[23,24]. In addition, conflicting scenarios have been proposed to explain the global spread of *T. gondii* populations[25–27], often with no distinction between ancient and recent histories. Given the crucial importance of domestic cats and rodents in the transmission of *T. gondii*, a presumed role of these host species in the recent global spread of the major clonal lineages has been repeatedly evoked in the literature[13,27–29]. However, this hypothesis could not be formally tested previously[27,29] owing to the paucity of *T. gondii* samples in many regions[24,25,28,30,31]. Moreover, a lack of good estimates of parasite mutation rate and generation time has hampered attempts to date dispersal time in relation to the expansion history of the principal hosts.

To address these questions, we focused our analyses on the recent evolutionary history of *T. gondii*, which is assumed to have been strongly influenced by migrations of strains between Old and New Worlds[27–29]. Using the largest dataset of *T. gondii* genomes produced to date (*n* = 156), we revisit the ancestry and admixture patterns of *T. gondii* populations worldwide. We provide a direct estimate of *T.*

*gondii* mutation rate and an estimate of its generation time, allowing us to date the major events that have shaped the parasite genome evolution. We use selection inference tools to identify the major selective processes acting on current *T. gondii* populations. We uncover candidate genes whose geographic distribution, genomic and stage-specific patterns of expression are most parsimoniously explained by a role in transmission by domestic cats.

## Results

Paired-end sequence reads from 59 publicly available haploid genomes and from 105 newly determined haploid genomes were aligned to the recent PacBio reference assembly RH-88. These samples included isolates from domestic (*n* = 107) and wild animals (*n* = 17) in addition to a number of human isolates (*n* = 40) (Supplementary Data 1). These samples originated from 22 countries (and three French Overseas Departments), and covered most of the global distribution of this species (Fig. 1). We had only a few samples from Asia (*n* = 3), but the predominant lineages found in the continent[14] were represented in our dataset (types I, II III, Chinese 1 and Africa 4). Note that most Old World samples included in this study belonged to clonal lineages previously identified and defined from MS markers (types I, II, III, Africa 1 and Africa 4; Supplementary Data 1), reflecting the very limited genetic diversity of *T. gondii* in the Old World. Many of these samples originated from port regions in Africa (Goree island, Saint-Louis, Dakar, Cotonou, Ouidah, Libreville) and Europe (Bordeaux, Le Havre), the most likely source populations for the recent human-mediated global expansion of *T. gondii* strains. Isolates from the New World originated from both coastal areas (e.g. Sao Paulo, Rio de Janeiro, Cayenne), and inland areas, including wild environments in South and North America. In addition, several isolates came from the Caribbean islands (Martinique, Guadeloupe), which are well-known for their importance in maritime history linking Old and New Worlds. New identifiers were assigned to strains included in this study, indicating the country of origin of each strain.

The 105 new haploid genomes included in this study were sequenced at a mean depth of 21X, ranging between 8 and 57X (Supplementary Data 1). In total, 156 genomes and 1,262,582 single-nucleotide polymorphisms (SNPs) passed all filtration criteria.

### Clonal lineages

We first wondered whether the clonal lineages identified in *T. gondii* based on the analyses of multilocus markers were evident at the genome-scale. Our analyses (see Supplementary Note 1 for details) revealed the presence of four intercontinental clonal lineages, as well as a number of regional (restricted to one continent) clonal lineages (Fig. 2). By matching these results to previous findings relying on the analysis of MS markers, we found that the four intercontinental clonal lineages correspond to type I (found in Asia, Europe, North and South America), type II (found on all continents), type III (found on all continents) and Africa 1 (found in Africa, South America and Western Asia)[12,14,16]. Africa 4 is found in both Africa and Asia, although samples of this lineage from Asia were not available for our study. Four South American clonal lineages could be identified, although these lineages were undersampled. Two corresponded to the previously described Caribbean 1 and Caribbean 2 lineages[18]. Most strains from South America were non-clonal.

### Global and local ancestry analyses

To decipher the genetic relationships between the *T. gondii* genomes from different geographical origins, we first performed ancestry analysis using unsupervised clustering with ADMIXTURE. This analysis was carried out after a step of clone-censoring of the dataset (keeping only one representative strain of each clonal lineage/type), resulting in a dataset of 71 strains and 588,777 SNPs. The optimal number of ancestral groups was determined to be five (lowest cross-validation

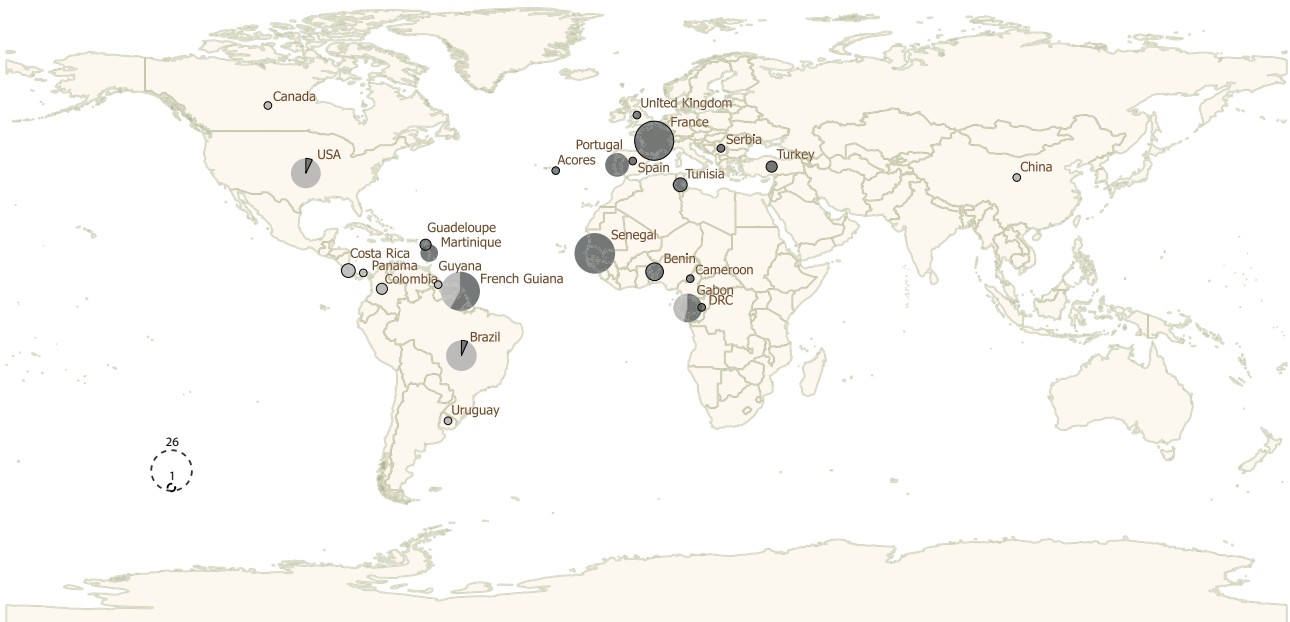

**Fig. 1 | The geographical distribution of *Toxoplasma gondii* strains analysed in this study.** Sizes of pie charts correlate with total number of mono-strains isolates for each country. Isolates sequenced specifically for this study are represented in dark grey and whole-genome sequence data from previous studies—publicly available on the European Nucleotide Archive (https://www.ebi.ac.uk/ena/browser/home)—are represented in light grey. DRC is an abbreviation of Democratic Republic of Congo.

error), but we also examined different K values. Old World strains not belonging to the predominant lineages (types I, II, III, Africa 1 and Africa 4) were found to be putative hybrids of these lineages (Supplementary Fig. 1). In the New World the Amazonian group in orange (composed of strains from the Amazonian forest) constituted a well-defined non-admixed ancestral group, clearly divergent from the different inter-continental clonal lineages at different K solutions. In addition, a group constituted of strains from North and South America isolated in the wild environment could be distinguished at K = 9, and was designated as Pan-American. Other New World lineages and strains were com-posed of New World-specific ancestries (in burgundy and in orange) and intercontinental lineage ancestries, with many of these strains exhibiting a mixed pattern of these two categories of ancestry in the same genome. This latter pattern was mainly noticed among strains from the domestic environment (urban and rural areas) and could evoke hybridization events between intercontinental lineages and New World-specific clades.

To better understand the patterns of admixture, we generated a co-ancestry matrix with whole nuclear genome data and independent co-ancestry matrices for each of the 13 nuclear chromosomes using ChromoPainter. This revealed that, at the exception of URUGUAY01, Pan-American and Amazonian populations, all New World strains shared chunks of chromosomes with at least one of the four inter-continental clonal lineages type I, type II, type III and Africa 1 (Supple-mentary Fig. 2a–n). In addition, many New World strains shared chunks of chromosomes with strains of Amazonian or Pan-American popula-tions, providing additional support for the hypothesis of their hybrid ancestry. By contrast, the two above-mentioned wild populations did not share any chromosome regions with the four intercontinental clonal lineages. Accordingly, we performed local ancestry analyses by defining the intercontinental lineages (type I, type II, type III, Africa 1 and Africa 4) and the New World non-hybrid wild populations (Amazonian and Pan-American) as putative ancestral populations. Putative hybrids (all the other genomes) were made up of a few large blocks of different ancestries (up to five ancestors for the same strain), and whole chro-mosomes of single ancestry were also often observed (Fig. 3). At least one large segment (>1 Mb) of ancestry corresponding to an intercontinental lineage was found in all putative hybrids from both Old and New Worlds. One exception was URUGUAY01 (CASTELLS) which had almost only an Amazonian-related ancestry. Amazonian-related ancestry was identified in nearly all putative hybrids from South and Central America, and to a lesser extent in North America, but was absent from almost all Old World strains. Pan-American ancestry was much rarer among putative hybrids, although it was found in both South and North America. Analyses of the apicoplast genome (a maternally inherited organelle that does not undergo recombination) confirmed the hybrid origin of those strains, as each apicoplast genome harboured a single ancestry, related either to an intercontinental lineage or to a New World wild population (Supplementary Fig. 3). In the Old Word, most hybrids were the result of an admixture between the major intercontinental lineages.

Overall, our results demonstrate that Old and New World *T. gondii* parasites present radically different patterns of genetic diversity. In the Old World (Europe, Africa and Asia), most strains belong to one of the previously defined intercontinental clonal lineages, with rare hybrids observed between the different lineages. On the other hand, most New world parasites isolated from wild environments form well-defined distinct populations, and most of those isolated from domestic environments are probably the result of hybridizations between intercontinental lineages and a number of New World-specific wild populations (Supplementary Fig. 4).

**Timing of emergence, spread and introgressions of inter-continental *T. gondii* clonal lineages**

Using stored aliquots of two long-term in vivo cultured *T. gondii*, we were able to estimate the average whole-genome mutation rate of the parasite (see supplementary Note 3). Based on these estimates, baye-sian phylogenetic analyses were carried out to estimate (1) the time to the most recent common ancestor (TMRCA) of each of the four major intercontinental clonal lineages, (2) divergence times between Old and New World strains of the same lineage and (3) between Old world and hybrid New World strains sharing the same ancestry at the level of chromosomes (see Methods). Lineage TMRCA was estimated at 791 (365–1294) years before present (YBP) for type I, 26,086 (11,290–47,143)

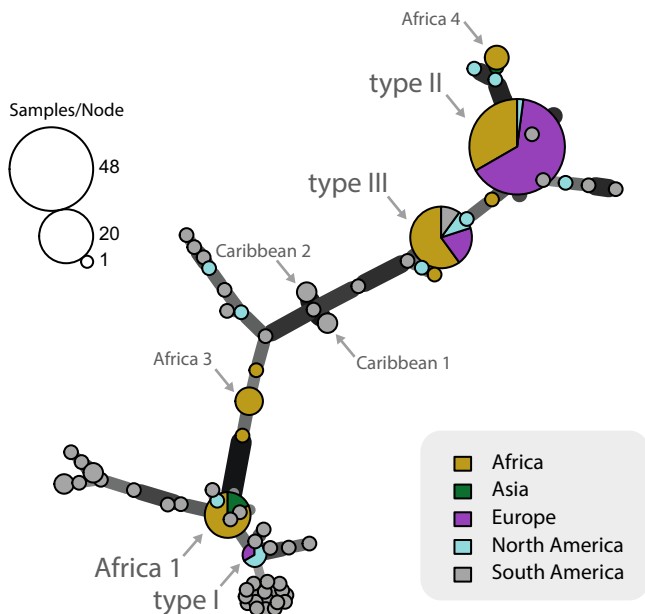

**Fig. 2 | *Toxoplasma gondii* clonal lineages description.** Minimum spanning network of *T. gondii* genomes. Genomes separated by a genetic distance less than or equal to 0.01 are collapsed in a single circle and are considered belonging to the same clonal lineage (See Supplementary Note 1). The size of each circle corresponds to the number of individuals, and the colours indicate the continent of origin of each individual. Thick and dark lines show MLGs that are more closely related to each other whereas edge length is arbitrary.

YBP for type II, 918 (419–1627) YBP for type III and 3993 (411–4697) YBP for Africa 1. However, strains of each of these lineages could not be sampled in many regions, possibly leading to an underestimation of true lineage emergence times (Supplementary Fig. 8). Old and New World strains (hybrid or not) of the same ancestry showed very low genetic divergence at whole-genome or chromosome level (Supplementary Figs. 5–6). Accordingly, estimated divergence times between Old and New World strains of the same ancestry at whole-genome and chromosome levels were very recent, with a median of 827 YBP (Q1:472; Q3:1,198) (Supplementary Fig. 7). Within each lineage, specific geographical populations involved in intercontinental migrations between Old and New Worlds are unlikely to have been optimally sampled in all regions. In consequence, strains analysed in this study likely exhibit a certain degree of divergence from these populations. This sampling bias probably resulted in an overestimation of divergence times between Old World and New World *T. gondii* populations, not forgetting that migration time is subsequent to divergence time (Supplementary Fig. 8). Overall, these TMRCA estimates show that massive migrations of strains took place between Old and New Worlds during the last few centuries. These migrations played a major role in shaping current *T. gondii* populations. The only exception to this pattern was observed among type 12 strains. Type 12 strains were shown to carry type II-related ancestry (Fig. 3) on certain chromosomes while exhibiting a relatively high degree of divergence from type II strains for these chromosomes (Supplementary Fig. 6). This pattern suggests that type 12 and type II ancestors diverged before the emergence of type II (Supplementary Discussion 1). Overall, phylogenies did not provide clear evidence for the direction of these migrations (from Old World to New World or vice versa).

### Candidate genes under selection in intercontinental lineages and hybrid populations

We asked whether hybrid strains from the New World (Fig. 3) are the result of random recombinations between intercontinental lineages and New World non-hybrid wild populations, or whether certain alleles

inherited from intercontinental lineages had been selected during this process. To this end, the 71 genomes of the clone-censored dataset were split into three groups based on results from global and local ancestry analyses: (1) intercontinental lineages and hybrid strains derived from intercontinental lineages, (2) New World non-hybrid wild populations (Amazonian, Pan-American and wild-type 12), and (3) URUGUAY01 (CASTELLS). This latter strain was used as an outgroup given its New World-specific ancestry (Fig. 3), and its divergence from other populations[24]. We computed the population branch statistic (PBS) in order to detect genomic regions of unexpectedly high divergence between the three groups. A unique outlier region of marked divergence between input groups was identified (Fig. 4a); it occurred on chromosome 1a over a region ~600 kb (Fig. 4b). In parallel, we scanned the nuclear genome for alleles common to intercontinental lineages and hybrid strains (group 1) not found in others populations (group 2). We chose to include URUGUAY01 in the second group, given its nearly exclusive Amazonian-related ancestry (Fig. 3).

In total, 310 variants were identified, of which 58 were missense variants (Supplementary Data 2). Strikingly, all 310 variants occurred within the outlier region identified using PBS statistic, over a region ~150 kb long containing 25 genes between positions 1,352,873 and 1,501,765 on chromosome 1a (Fig. 4c). When we considered the ancestry pattern of this region (Fig. 3), we noticed that nearly all hybrid strains have a type II/type III ancestry between positions 1,390,579 and 1,502,589 (exact boundaries were extracted from the output files of Ancestry_HMM). This latter genomic region of ~100 kb on chr01a was shown to contain 21 genes (Supplementary Data 3). The phylogeny revealed that this region divides the global dataset into two clades: the first clade comprises intercontinental lineages and all hybrid strains, while the second comprises Amazonian, Pan-American, wild-type 12 and URUGUAY01 (Fig. 4d). Most Old and New World strains composing the first clade shared a remarkably conserved haplotype at this genomic region. Basal positions in this clade were occupied by Old World lineages (Africa 4 and Chinese 1), enabling to identify the Old World as the most plausible origin of this clade. Extended linkage disequilibrium was observed around the outlier region in the first clade relative to the second, a sign of positive selection acting specifically on intercontinental lineages and hybrid strains (Supplementary Fig. 9).

In order to determine which gene(s) is/are most likely concerned with the selective process acting on the outlier region, we carefully analysed the predicted function of the 21 candidate genes present in the ~100 kb genomic region[32,33]. Although 10 genes were annotated with a putative protein function, possible associations between allelic heterogeneity and differential adaptations to specific hosts or ecotypes were not reported[32,33]. Therefore candidate genes were ranked according to three criteria: (1) presence of missense (nonsynonymous) variants specific to intercontinental lineages and hybrids; (2) expression, increased expression, or specific expression during enteroepithelial stages (EES) of development (early cellular forms characteristic of the onset of the sexual stage in cat enterocytes); and (3) degree of conservation in comparison to orthologs from the closest species to *T. gondii* (*Hammondia* or *Neospora*). Sixteen genes were considered to be poor candidates as they did not carry missense variants, were not expressed, showed a significant decrease in their expression during EES, or were highly conserved (>95%). Among the five remaining candidate genes, only one was found to be specifically expressed during EES: TGRH88_020330 (TGME49_295920) and is annotated as encoding a hypothetical protein. Five missense variants specific to intercontinental lineages and hybrids were found for this gene (Supplementary Table 1).

## Discussion

We showed that intercontinental *T. gondii* lineages have recently hybridized with a number of New World populations, leading to the emergence of numerous hybrid *T. gondii* populations in the New

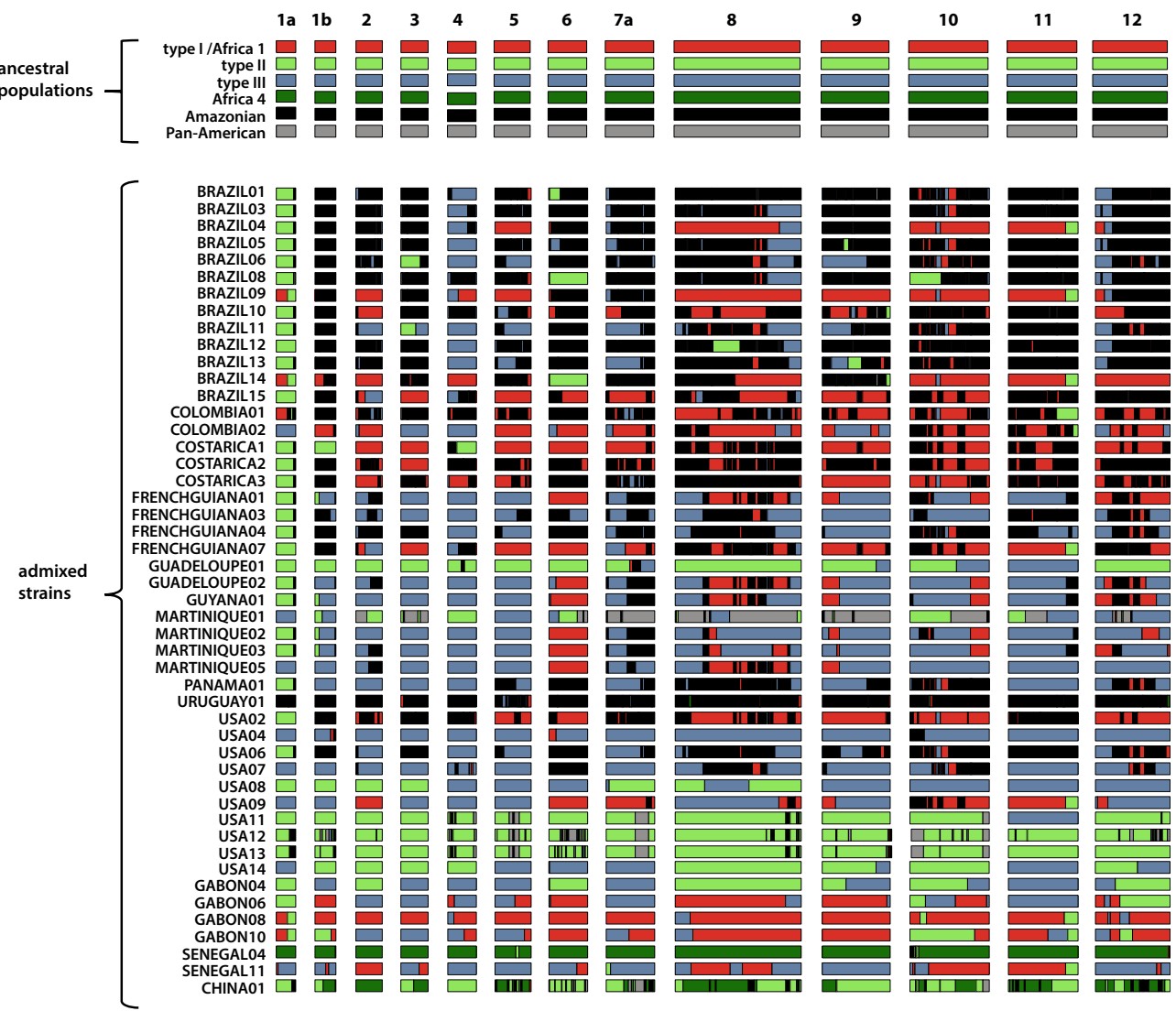

**Fig. 3 | Genome-wide distribution of ancestry in all putative hybrid *Toxoplasma gondii* genomes.** Plots are graphically displayed using karyoploteR[68] and show ancestry estimates at each genomic position for the 13 nuclear chromosomes (1a–12). Colours in plots reflect putative ancestral populations.

World. We identified a unique haplotype of ~100 kb (0.16% of the whole *T. gondii* genome) on chromosome 1a common to all intercontinental *T. gondii* lineages and hybrid strains and under strong positive selection in these populations. On the opposite, New World non-hybrid populations carried a diversity of divergent haplotypes in this genomic region. Note that a number of intercontinental *T. gondii* lineages and hybrid strains share the same haplotype for the whole length of chromosome 1a, a pattern previously noticed in past studies[24,31]. Our results indicate that this pattern can be explained by the much stronger linkage disequilibrium (often reaching the whole length of chromosome 1a) observed in these strains relative to New World non-hybrid populations around the ~100 kb outlier region, which probably persists due to the combined effects of selection and rarity of sexual recombinations in most *T. gondii* populations.

All *T. gondii* strains from the Old World carried the ~100 kb unique haplotype of *T. gondii* on chromosome 1a, whether they were isolated in a domestic or wild environment[13,14,34]. Almost all *T. gondii* strains from the New World carried the same haplotype when they were isolated in the domestic environment. However, this haplotype showed various degrees of dissemination to wildlife in North and South America according to geographic area[17,18,35] (Supplementary Discussion 2). By contrast, *T. gondii* strains from the New World carrying divergent haplotypes appeared to be restricted to the wild

environment. At a global scale, the environmental distribution of the ~100 kb unique haplotype appears to be a reflection of the dissemination pattern of oocysts shed by domestic cats[36]. Indeed, the domestic cat is virtually the only host capable of disseminating oocysts in the domestic environment. In most regions of the Old World, populations of wild felids have undergone a massive decline in numbers[37–41], leaving domestic cats as virtually the only shedders of oocysts, which spread over longer distances via waterways to reach wildlife[42,43] (also refer to Supplementary Discussion 3). A comparable situation has been described in the New World, with the exception that large populations of wild felids are still observed in certain regions of North and South America (Supplementary Discussion 2), and this is where *T. gondii* strains with divergent haplotypes on chromosome 1a are found.

Phylogeny supported the hypothesis of an Old World origin of the ~100 kb unique haplotype. This haplotype was found to be carried by the four major intercontinental lineages (types I, II, III and Africa 1). In consequence, the most straightforward scenario would be that these lineages originated in the Old World and subsequently seeded to the New World, although other scenarios could be considered (Supplementary Discussion 3). In this scenario, this haplotype would have followed the same route as domestic cats (*Felis catus*), which also emerged in the Old World from wildcats (*Felis silvestris*) before

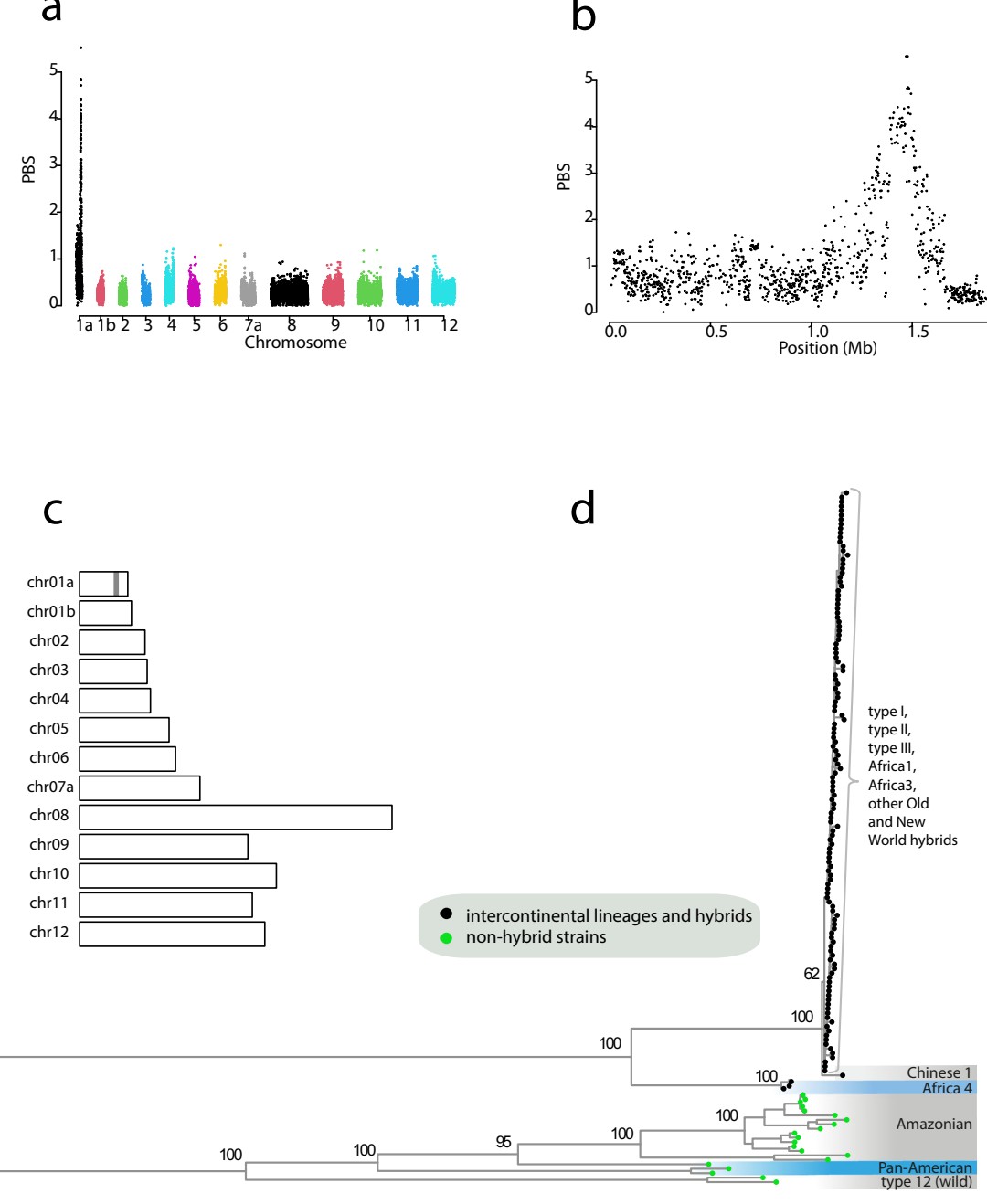

**Fig. 4 | Outlier region under selection in intercontinental lineages and hybrid strains. a** The Manhattan plot shows the genome-wide distribution of population branch statistic (PBS) for each 50-SNP sliding window using the clone-censored dataset ($n = 71$). **b** This is a zoom on PBS obtained for chromosome 1a. **c** Genomic positions of variants specific to intercontinental lineages and hybrid strains ($n = 310$) from the clone-censored dataset are indicated by grey bars. **d** Phylogenetic tree of the outlier selection region (positions 1,390,579–1,502,589 on chromosome 1a). It was produced using iqtree2 under General Time-Reversible (GTR) nucleotide substitution model with a gamma distribution and a proportion of invariant sites (GTR + Γ + I). Branch support was inferred using 1000 bootstrap replicates. It includes all the strains of the dataset ($n = 156$).

reaching the New World in the last few centuries[44–46]. This would support a temporal association between this haplotype and the presence of cats (Fig. 5), in consistency with their current geographical association as described above.

It is noteworthy that this haplotype was circulating in the Old World well before these recent times and even before cat domestication (10,000 years ago), according to phylogeny. For example, type II, one of the lineages carrying this haplotype, was estimated to have emerged 26,000 years before the present. Therefore, it is likely

that this haplotype was already circulating in wildcats (*Felis silvestris*) before domestication. The domestication of cats first occurred in the Near East coincident with agricultural village development in the Fertile Crescent about 10,000 years ago[44,46]. Since this period, regular genetic exchanges occurred between domesticated cats (*Felis catus*) and their wild counterparts (*Felis silvestris*), probably contributing to the pattern of low level of differentiation observed between modern wild and domestic cat genome sequences[47]. It is therefore likely that *T. gondii* populations transmitted by wildcats

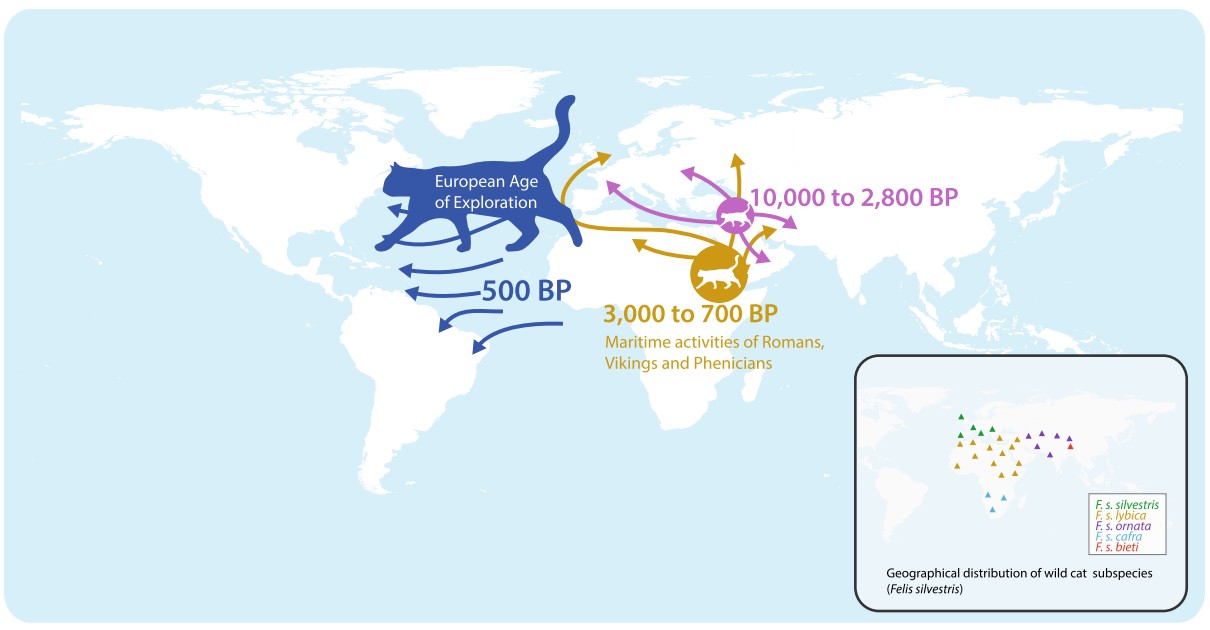

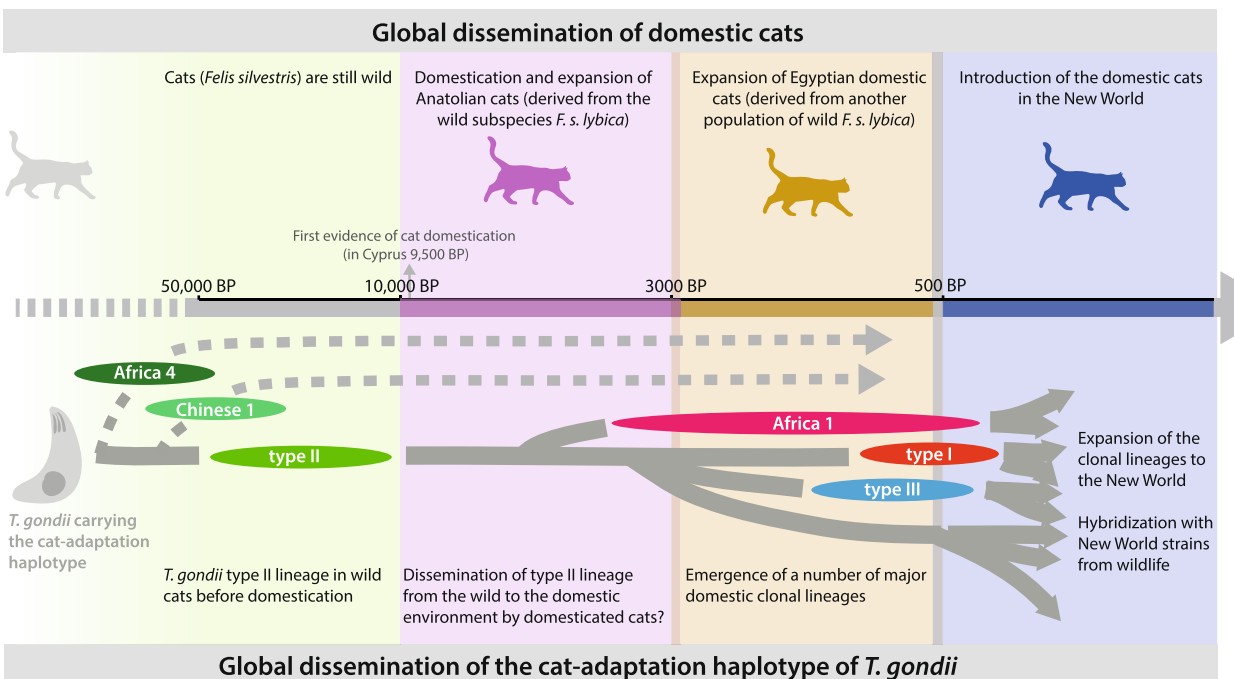

**Fig. 5 | Graphical summary of major events that have punctuated the recent evolutionary history of *Toxoplasma gondii* in relation to history of cats' dispersal.** Geographical distribution of wildcat subspecies and historical data about cats' dispersal are derived from[46]. The grey dotted arrows in the bottom part of the figure indicates that emergence of Africa 4 and Chinese 1 lineages could not be dated in this study due to a lack of samples of these two lineages. BP before present.

did not need further adaptation to occupy the new niches offered by domesticated cats, as shown by the marked genetic conservation of the -100 kb haplotype from the emergence of type II until today. *T. gondii* strains carrying this haplotype could have accompanied cats of subspecies *F. s. lybica* (one of the five known clades of wildcats of species *Felis silvestris* found in different regions of the Old World) that established themselves in human settlements in the Near East, or emerged later in the domestic environment from other wildcat populations in Africa, Asia or Europe following the expansion of the Neolithic revolution to those regions. At least three genetic populations of this haplotype were at the origin of its spread in the domestic environment: Africa 4, which expanded in Africa and Asia, Chinese 1 found today in Eastern Asia and type II, at the origin of its global expansion.

Type I, type III and Africa 1 lineages appear to have emerged much later, although TMRCA estimates support that they most probably emerged before the dissemination of domestic cats to the New World 500 years ago; it's hence likely that they emerged in the Old World (Supplementary Discussion 3–4). Note that the inclusion of additional isolates from the same or different geographic areas could alter these estimates by revealing more ancient divergence times between strains of each respective lineage. This is particularly true for type I for which only three samples were available.

TMRCA estimates indicate that hybridizations between intercontinental lineages and New World non-hybrid populations occurred in the last few centuries and coincide with the onset of the European "age of exploration"[48]. During this period, human activities enabled domestic cats to reach the New World for the first time[45] allowing the

emergence of a domestic life cycle of *T. gondii* in those areas. Importantly, the ~100 kb unique haplotype of *T. gondii* was under strong positive selection during these hybridizations, since it was inherited by all New World hybrids, probably from one of the major lineages (types I, II, III or Africa 1). It is noteworthy that hybrid strains are more diversified in South America compared to the rest of the World, indicating more frequent recombination events, likely due to the effect of superinfections[49,50] (Supplementary Discussion 5). Overall, these findings further illustrate the tight association between the expansion of cats and the dissemination and the selection of the ~100 kb unique haplotype of *T. gondii*.

A *T. gondii* strain is transmitted by the definitive host only if it undergoes sexual reproduction in enterocytes, and other potential routes of transmission were previously shown to be inefficient in cats[51,52]. Therefore, the selection of the ~100 kb unique haplotype on chromosome 1a is most parsimoniously explained by its role in the efficiency of sexual reproduction of *T. gondii* in cats. This is consistent with results from a previous study by Khan et al.[22], which showed that *T. gondii* strains from the domestic environment of French Guiana (Caribbean lineages) are probably more efficient at being transmitted in the form of oocysts by cats than wild strains from the same area. This should be confirmed on larger sample sizes, but the study of *T. gondii* sexual reproduction is hampered by the important ethical concerns associated with the use of live cats.

In the present study, we identified five genes within the ~100 kb genomic region on chromosome 1a which fulfil the criteria of promising candidate genes due to their possible involvement in sexual reproduction in cat enterocytes[32,33]. A selective process taking place during host cell recognition or invasion before the initiation of sexual reproduction should also be a hypothesis to consider. In this case, other genes on the ~100 kb outlier region of chromosome 1a could merit consideration despite not being expressed during the different phases of sexual reproduction *stricto sensu*. TGRH88_020330 (TGME49_295920) was selected as the top candidate gene, as it was the only gene specifically expressed by *T. gondii* during the early stages of sexual reproduction in cat enterocytes, which also carried variants of putative functional relevance (missense) segregating the ~100 kb unique haplotype from divergent ones. However, it was not possible to determine which of these missense variants has functional relevance. The genomic region carrying this gene exhibits strong linkage disequilibrium among strains carrying the ~100 kb unique haplotype, implying that most variants associated with this haplotype ($n = 310$) have been fixed by hitch-hiking. Di Genova et al.[53] have recently produced cat intestinal organoids for experimental purposes, an important breakthrough to study the sexual reproduction of *T. gondii* and testing the function of the most promising candidate genes identified in this study. In the same way, developing similar experimental models from intestinal cells of wild felids could enable a more comprehensive understanding of the function of these genes.

In summary, we have produced a large dataset of high-quality *T. gondii* genomes and provided estimates of the parasite's mutation rate and generation time. We show that a ~100 kb unique haplotype on chromosome 1a has recently spread worldwide. It has been largely conserved since its initial emergence from wilderness to domestic settlements in the Old World, its dissemination, and its recent expansion to the New World. Our results argue that the selection of this haplotype can be explained by its role in augmenting the efficiency of sexual reproduction of *T. gondii* in cats. This haplotype could be conferring a selective advantage to strains carrying it in the domestic environment and could be one of the key adaptations at the origin of the global spread of this parasite, estimated to be infecting one-third of the world's human population[2]. Our findings constitute an example of the profound effect exerted by human activities on pathogen diversity brought about by two major events in human history: domestication during the Neolithic revolution and globalization of trade in the few last centuries.

## Methods

### Parasite strains

We studied 106 *T. gondii* isolates provided by the French Biological Resource Centre (BRC) for *Toxoplasma* (http://www.toxocrb.com/). This certified structure (NF S96-900 standard) manages the storage of *T. gondii* strains from human or animal toxoplasmosis to make them available to the scientific community. Our analyses were complemented with whole-genome sequence data of 59 strains (100 bp paired-end reads) from a previous study[24] made publicly available on the European Nucleotide Archive (https://www.ebi.ac.uk/ena/browser/home). Following parasite culture (Supplementary Note 2), total genomic DNA was extracted from 200 μl of tachyzoites suspension, using the QIAamp DNA MiniKit (Qiagen, Courtaboeuf, France). *Toxoplasma gondii* DNA extracts were genotyped using 15 MS markers (Supplementary Data 4) in a single multiplex PCR-assay[54]. Briefly, the forward primer from each pair was 5′-end labelled with fluorescein as follow: 6-carboxyfluorescein (6-FAM) was used for TUB2, XI.1, B18, N83, N61, M33 and M48; hexachlorofluorescein (HEX) for MS TgM-A, B17, N82, W35 and IV.1; and 2,7′,8-benzo-5′-fluoro-2,4,7-trichloro-5-carboxyfluorescein (NED) for AA, N60 and M102. The PCR reaction was carried out using a 25 μl reaction mixture composed of 12.5 μl of 2× Qiagen Multiplex PCR Master Mix (Qiagen, Courtaboeuf, France), 5 pmol of each primer and 5 μl DNA. Cycling conditions were as follows: initial denaturation 15 min at 95 °C, followed by 35 cycles of 94 °C for 30 s, 61 °C for 3 min, 72 °C for 30 s, and 30 min at 60 °C. PCR products were diluted 1:10 in deionized formamide (Applied Biosystems, Life Technologies, Carlsbad, California). One microliter of each diluted PCR product was mixed with 0.5 μl of a dye-labelled size standard (ROX 500, Applied Biosystems) and 23.5 μl of deionized formamide (Applied Biosystems). This mixture was denatured at 95 °C for 5 min. The PCR products were then electrophoresed using an automatic sequencer (ABI PRISM 3130xl, Applied Biosystems). The size of the alleles in bp was estimated using GeneMapper analysis software (version 5.0, Applied Biosystems). This genotyping step was necessary to check for cross-contaminations between samples during culture and to identify mixed infections. Mixed infection was identified in one isolate (FR-Mac fas-002; *Macaca fascicularis*; Mauritius) by the presence of two alleles at 12 loci; only one strain was genotyped at the time of isolation, and it is therefore likely that the second strain initially had a lower tissue load in the infected host, and took longer to grow to levels sufficient for library construction. Only the mono-strain isolates ($n = 105$) were sequenced. DNA was sheared into 400–600-base pair fragments by focused ultrasonication (Covaris Adaptive Focused Acoustics technology, AFA Inc, Woburn, USA). Standard indexed Illumina libraries were prepared using the NEBNext DNA Library Prep kit (New England BioLabs), followed by amplification using KAPA HiFI DNA polymerase (KAPA Biosystems). 150 bp paired-end reads were generated on the Illumina NextSeq 500 according to the manufacturer's standard sequencing protocol.

### Mapping and variant calling

Recent advances in variant calling enabled the use of publicly available panels of validated single-nucleotide polymorphisms (SNPs) and indels to estimate the accuracy of each base call and minimize the generation of false positive SNPs. Unfortunately, building these panels is labour-intensive and such data is lacking for under-studied organisms such as *T. gondii*, which still fall under the category of "non-model" organisms. Ribeiro et al.[55] explored the relationship between the choice of tools and parameters in non-model organisms, and their impact on false positive variants, and formulated recommendations for variant calling. Here, we followed their recommendations in order to minimize the call of false positive variants, which was a critical point regarding our objective of estimating the occurrence times of recent events in *T. gondii* evolution. In this sense, reads were submitted to a stringent mapping configuration (not more than 2% of mismatches), by using BWA 0.7 against the newly available PacBio reference genome RH-88 (13 nuclear

chromosomes that cover 63.97 Mb; GCA_013099955.1; release date 2020-05-15). Mapped reads were sorted with Samtools 1.11, and duplicate reads were marked with Picard 'MarkDuplicates' 2.25. Individual BAMs were subsequently merged, and variant calling was performed with FreeBayes 1.3.5[56], which is considered better than the routinely used Genome Analysis Toolkit (GATK) in non-model organisms[55,57]. At the individual level, alignments having a mapping quality (--min-mapping-quality) less than 20, a coverage (--min-coverage) less than 3, and alleles having a supporting base quality (--min-base-quality) less than 20 were excluded from the analysis. Genotype calls having a fraction of conflicting base calls of more than 10% were also excluded. Finally, only individuals having missing genotype data of less than 5% were kept for subsequent analyses to minimize false negative calls. At the population level, singletons SNPs and SNPs having a high missing genotype rate (>10%) were filtered out. With the above filters in place, two individuals were excluded (missing genotype data >5%), before excluding an additional six samples due to unreliable information about the country of origin and/or the ecotype. The filtered set of SNPs was annotated with snpEff 5.0 6 using the RH-88 annotation file.

For the 35 kb *T. gondii* apicoplast genome, mapping of reads was performed against the ME49 reference genome assembly (GCA_000006565.2; release date 2013-04-23), as the PacBio reference genome, RH-88 had a high proportion of low complexity sequence (~70% in RH-88 genome versus ~30% in ME49 reference genome). The mapping quality of reads in these low complexity regions was too low, and therefore these regions were not exploitable for sequence analysis. A mapping configuration and variant calling parameters identical to those used for the nuclear genome were used for the apicoplast genome. In addition to the six samples having unreliable information about the country of origin and/or the ecotype, six additional samples were excluded due to a high frequency of missing genotype data (>5%).

## Clonality

Most computational tools for population genetics are based on concepts developed for sexual model organisms. Microbial pathogens are often clonal or partially clonal, and hence require different tools to address their population dynamics and evolutionary history. The R package poppr 2.0[58] specifically addresses issues with the analysis of clonal and partially clonal populations. We first used this package to collapse individuals into clonal groups, by defining a genetic distance threshold based on 3 different clustering algorithms using the function mlg.filter[58]. This initial step, besides enabling the definition of clonal lineage boundaries, is a necessary partial correction for a bias that affects metrics of most computational tools that often rely on allele frequencies assuming panmixia. A dissimilarity matrix was produced by poppr to compute genetic distances between genomes, and a minimum spanning network was drawn based on these calculations, by collapsing individuals based on the previously defined genetic distance threshold.

## Global ancestry inference

In order to identify ancestral populations and characterize the admixture patterns in our dataset, we used ADMIXTURE 1.3[59]. ADMIXTURE is useful as an exploratory tool in analyses of genetic structure but should be interpreted with caution since such model-based algorithms often provide only a caricature of a complex reality. The dataset was clone-censored (including only one randomly chosen strain from each clonal lineage) and pruned for linkage disequilibrium in PLINK[60] (v. 1.07) using parameters --indep-pairwise 50 5 0.2 (it removes each SNP that has an $R^2$ value greater than 0.2 with any other SNP within a 50-SNP sliding window, advanced by 5 SNPs each time). ADMIXTURE was run using the unsupervised mode with cross-validation (--cv). The number of ancestral populations (K) varied between 2 and 10.

We complemented our global ancestry analyses with ChromoPainter[61], known to be particularly useful to discern signatures of recent admixture. ChromoPainter estimates the number of "chunks"

of ancestry inherited by an individual or a population from a "donor" individual or population and builds a co-ancestry matrix that summarizes the degree of sharing of ancestry among all pairs of individuals. Unlike ADMIXTURE, ChromoPainter takes into account patterns of linkage disequilibrium, allowing one to combine information across successive markers to increase the ability to capture fine-scale population structure. Hence, linkage disequilibrium pruning was not required for this analysis, and we used the unpruned clone-censored dataset.

## Local ancestry inference

Local ancestry inference (also designated as ancestry deconvolution) is the task of identifying the regional ancestral origin of chromosomal segments in admixed individuals. It requires specifying a set of candidate non-admixed populations as putative ancestors of the admixed individuals. After defining ancestral and admixed individuals using approaches of global ancestry inference, we carried out local ancestry analysis using the recently released Ancestry_HMM software[62]. Ancestry_HMM is based on a novel hidden Markov model that does not require genotypes from reference panels and that is generalized to arbitrary ploidy, and is hence suitable for non-model haploid organisms. In addition, a TCS network was produced from the apicoplast sequences using PopART.

## Dating emergence, global spread and introgressions of intercontinental *T. gondii* clonal lineages

We estimated the mutation rate of *T. gondii* based on the in vivo mutation rate of the RH strain, through the successive passage in outbred mice for 30 years (Supplementary Note 3). The estimated mutation rate ranged between $3.1 \times 10^{-9}$ to $11.7 \times 10^{-9}$ (mean: $7.4 \times 10^{-9}$) mutations per site per year. This mutation rate was used for dating the emergence of intercontinental lineages and for estimating the migration times of *T. gondii* populations between Old and New Worlds. These dating analyses were carried out using the Bayesian Markov chain Monte Carlo (MCMC) approach implemented in BEAST v1.10.4. The General Time-Reversible (GTR) nucleotide substitution model with a gamma distribution and a proportion of invariant sites (GTR + Γ + I) was identified as the best fitting model by jModelTest v2.1.10. The Markov Chain Monte Carlo (MCMC) was run for 100 million steps with sampling parameters and trees every 10,000 steps. The convergence of all parameters was assessed by calculating effective sample sizes (ESS), setting their threshold at 200 after discarding the initial 10 per cent of each run as a burn-in using Tracer v1.7.1. By using the above-mentioned model configuration, genomic sequence datasets were produced using Bcftools consensus 1.11. A genotype call falling below minimal quality requirements (--min-mapping-quality 20; --min-coverage 3; --min-base-quality 20; conflicting base calls <10%) was considered as a low-quality genotype and was masked with an N character to symbolize that the nucleotide base at this position is unknown.

For genomes of each intercontinental lineage, a specific dataset was produced, and an independent run was launched to estimate their TMRCA as an approximation of lineage emergence time. We assumed a strict molecular clock in model configuration since the genomic sequences used in each run were all from the same lineage. Clock rate was set to $7.4 \times 10^{-9}$ and the standard deviation to $2.15 \times 10^{-9}$. When strains of a given intercontinental lineage shared the same ancestry with other strains at a given chromosome (according to local ancestry analysis), their genetic closeness was confirmed by building a chromosome-specific neighbour-joining tree using ape R package that computed genetic distances based on the dissimilarity matrix produced by poppr R package. Following this verification step, a specific dataset was produced for the chromosome sequences of these two groups to estimate their divergence times. Independent runs were launched for each specific dataset in BEAST under an uncorrelated lognormal relaxed clock. Uncorrelated relaxed clocks allow each branch of a phylogenetic tree to have its own evolutionary rate[63] to account for possible variability in evolutionary rates within the tree.

The mean rate of the molecular clock (ucld.mean) was set to $7.4 \times 10^{-9}$ and the standard deviation (ucld.stdev) to $2.15 \times 10^{-9}$. In order to estimate divergence times between Old and New Worlds *T. gondii* populations, the TMRCA of each New World strain with its closest Old World relative was extracted from the phylogenetic trees generated with BEAST using FigTree 1.4.4. All these TMRCA values were used to generate a boxplot to graphically visualize their distribution.

## Candidate genes for the adaptation to cats

To identify candidate genes involved in the process of adaptation to the domestic environment, a multistage process was used. We first carried out a divergence-based selection scan using the population branch statistic (PBS) software PBScan[64]. This method enables to detection of genomic regions of unexpectedly high differentiation between different pre-defined groups (estimated with FST measure by Hudson et al.[65]), a pattern indicative of directional selection[66]. In parallel, in order to ascertain candidate SNPs, we used the bcftools 1.1 -- *private* function to identify SNPs common to intercontinental linages and hybrid strains derived from these lineages. SNPs identified with these two approaches were annotated with SnpEff in order to define a primary list of candidate genes. To determine which candidate genes are most likely to be under positive selection, we focused on functional evidence. We carefully searched the literature for the function of candidate genes and their stage-related patterns of expression. Selection can result in patterns of extended linkage disequilibrium (LD) and extended haplotype homozygosity (EHH) around the selected site, especially relative to the alternative allele. EHH is defined as the probability that two randomly chosen chromosomes carrying the same allele at a focal SNP are identical by descent over a given distance surrounding it. Hence, for outlier SNPs validated by all types of evidence—environmental, statistical and functional—we computed EHH using the REHH 2.0 R package[67] to examine the extent of linkage disequilibrium around them.

## Reporting summary

Further information on research design is available in the Nature Research Reporting Summary linked to this article.

# Data availability

The 105 genomes sequenced in this study have been deposited in the ENA database under accession code BENIN01 (ERS13421591); BENIN02 (ERS13421592); BENIN04 (ERS13421593); BENIN05 (ERS13421594); BENIN03 (ERS13421595); BRAZIL01 (ERS13421596); CAMEROON01 (ERS13421597); DROC01 (ERS13421598); FRANCE01(ERS13421599); FRANCE02 (ERS13421600); FRANCE03 (ERS13421601); FRANCE04 (ERS13421602); FRANCE05 (ERS13421603); FRANCE06 (ERS13421604); FRANCE07 (ERS13421605); FRANCE08 (ERS13421606); FRANCE09(ERS13421607); FRANCE10 (ERS13421608); FRANCE11 (ERS13421609); FRANCE12 (ERS13421610); FRANCE13 (ERS13421611); FRANCE14 (ERS13421612); FRANCE15 (ERS13421613); FRANCE16 (ERS13421614); FRANCE17(ERS13421615); FRANCE18 (ERS13421616); FRANCE19 (ERS13421617); FRANCE21 (ERS13421618); FRANCE22 (ERS13421619); FRANCE23 (ERS13421620); FRANCE24 (ERS13421621); FRANCE20 (ERS13421622); FRENCHGUIANA08 (ERS13421623); FRENCHGUIANA09 (ERS13421624); FRENCHGUIANA11 (ERS13421625); FRENCHGUIANA13 (ERS13421626); FRENCHGUIANA14 (ERS13421627); FRENCHGUIANA15 (ERS13421628); FRENCHGUIANA01(ERS13421629); FRENCHGUIANA02 (ERS13421630); FRENCHGUIANA05 (ERS13421631); FRENCHGUIANA06 (ERS13421632); FRENCHGUIANA03 (ERS13421633); FRENCHGUIANA04 (ERS13421634); FRENCHGUIANA10 (ERS13421635); FRENCHGUIANA07 (ERS13421636); GABON01 (ERS13421637); GABON03 (ERS13421638); GABON05 (ERS13421639); GABON07 (ERS13421640); GABON02 (ERS13421641); GABON04 (ERS13421642); GABON06 (ERS13421643); GUADELOUPE02 (ERS13421644); GUADELOUPE01 (ERS13421645); MARTINIQUE04 (ERS13421646); MARTINIQUE02 (ERS13421647); MARTINIQUE03 (ERS13421648); MARTINIQUE01 (ERS13421649); MARTINIQUE05 (ERS13421650); PORTUGAL01 (ERS13421651); PORTUGAL03 (ERS13421652); PORTUGAL05 (ERS13421653); PORTUGAL06 (ERS13421654); PORTUGAL07 (ERS13421655); PORTUGAL08 (ERS13421656); PORTUGAL02 (ERS13421657); PORTUGAL04 (ERS13421658); PORTUGAL09 (ERS13421659); PORTUGAL10 (ERS13421660); SENEGAL05 (ERS13421661); SENEGAL10 (ERS13421662); SENEGAL22 (ERS13421663); SENEGAL25 (ERS13421664); SENEGAL04 (ERS13421665);SENEGAL14 (ERS13421666); SENEGAL23 (ERS13421667); SENEGAL01 (ERS13421668); SENEGAL02 (ERS13421669); SENEGAL03 (ERS13421670); SENEGAL06 (ERS13421671); SENEGAL07 (ERS13421672); SENEGAL08(ERS13421673); SENEGAL12 (ERS13421674); SENEGAL13 (ERS13421675); SENEGAL16 (ERS13421676); SENEGAL17 (ERS13421677); SENEGAL18 (ERS13421678); SENEGAL19 (ERS13421679); SENEGAL21 (ERS13421680);SENEGAL24 (ERS13421681); SENEGAL09 (ERS13421682); SENEGAL15 (ERS13421683); SENEGAL20 (ERS13421684); SENEGAL26 (ERS13421685); SENEGAL11 (ERS13421686); SERBIA01 (ERS13421687); SPAIN01(ERS13421688); TUNISIA01 (ERS13421689); TUNISIA03 (ERS13421690); TUNISIA02 (ERS13421691); TURKEY01 (ERS13421692); TURKEY02 (ERS13421693); UK01 (ERS13421694); USA01 (ERS13421695). RH2019 (ERR10079368); RH1989 (ERR10079367); PRU2019 (ERR10079366); PRU1989 (ERR10079365) were sequenced for datation purposes. RH-88 (GCA_013099955.1) and ME49 (GCA_000006565.2) were used as reference genomes in this study. The following publicly available sequence reads were also used in this study: BRAZIL02,TgCatBr01 (SRX099790); BRAZIL03,TgCatBr05 (SRX099804 SRX099805 SRX099795); BRAZIL04,TgCatBr09 (SRX099807 SRX099781 SRX099806); BRAZIL05,TgCatBr10 (SRX099791); BRAZIL06,TgCatBr15 (SRX099779); BRAZIL07,TgCatBr18 (SRX099794); BRAZIL08,TgCatBr25 (SRX160134); BRAZIL09,TgCatBr26 (SRX099780); BRAZIL10,TgCatBr3 (SRX160142); BRAZIL11,TgCatBr34 (SRX160051); BRAZIL12,TgCatBr44 (SRX160141); BRAZIL13,TgCatBr64 (SRX160743); BRAZIL14,TgCatBr72 (SRX160049); BRAZIL15,TgCkBr141 (SRX160124); CANADA01,COUG; TgCgCa01 (SRX099803 SRX099786 SRX099802); CHINA01,PgCatPRC2 (SRX156168 SRX155517); COLOMBIA01,TgCtCo5 (SRX156192 SRX156164 SRX156155 SRX154747); COLOMBIA02,TgDgCo17 (SRX099787); COSTARICA1,TgCkCr01 (SRX160807); COSTARICA2,TgCkCr10 (SRX099784); COSTARICA3,TgRsCr01 (SRX160143); FRENCHGUIANA16,GUY-DOS (SRX099782); FRENCHGUIANA17,GUY-KOE (SRX099796); FRENCHGUIANA18,GUY-MAT (SRX099783); FRENCHGUIANA19,GUY-2003-MEL (SRX160131); FRENCHGUIANA20,GUY-2004-ABE (SRX160132); FRENCHGUIANA21,GUY009-AKO (SRX171132); FRENCHGUIANA22,GUY 021−TOJ (SRX160041); FRENCHGUIANA24,RUB; GUY-RUB (SRX055419 SRX055414 SRX099773 SRX055412); FRENCHGUIANA12,VAND; GUY-VAND (SRX055413 SRX038726 SRX055418 SRX055416); FRENCHGUIANA23,GUY-JAG1 (SRX099776); GABON08,GAB3-2007-GAL-DOM2 (SRX160123); GABON09,GAB5-2007-GAL-DOM1 (SRX160069); GABON10,GAB1-2007-GAL-DOM10 (SRX159841 SRX159839); GABON11,GAB2-2007-GAL-DOM2 (SRX156037 SRX155963 SRX155534); GABON12,GAB3-2007-GAL-DOM9 (SRX160125); GABON13,GAB5-2007-GAL-DOM6 (SRX159842 SRX159840); GUYANA01,TgCkGy2 (SRX099785); PANAMA01,G662M (SRX160052); BOF,BOF; BE-BOF (SRX099774); FOU,FOU; BRE-FOU (SRX046277 SRX038725 SRX046278); LGE-CUV,LGE-CUV (SRX099775); MAS,MAS; LPN-MAS (SRX057823 SRX038728 SRX038699); TgH20005,IPP-URB (SRX157499 SRX157465); TgH26044 (SRX160050); URUGUAY01,CASTELLS (SRX099789); USA12,ARI (SRX099777 SRX099800 SRX099801); USA11,B41 (SRX099774); USA13,RAY (SRX099793); USA03,GT1 (SRX156314); USA05,ME49 (SRX055421 SRX055417 SRX055410); USA04,M7741 (SRX159890 SRX159849); USA10,VEG (SRX156300); USA14,B73 (SRX159844 SRX159843); USA02,CAST; US-CAST (SRX099788); USA06,P89; TgPgUs15 (SRX038693 SRX055420 SRX038727); USA07,ROD; ROD-US (SRX160129 SRX160128); USA08,SOU (SRX160119); USA09,TgShUs28 (SRX160130).

## Code availability

All the codes for the analyses presented in this paper, including the analysis pipeline, is described in detail in Methods and is available in published papers and public websites or, for in-house pipelines, is available upon reasonable request from the corresponding author.

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

## Acknowledgements

We thank the French Agence Nationale de la Recherche (ANR project IntroTox 17-CE35-0004) and the Nouvelle-Aquitaine region (Directorate of Research, Higher Education and Technology Transfer) for providing research funds to A.M. We thank Nicolas Plault, Lionel Forestier and Eden Lebrault for technical assistance. We want to acknowledge the Biological Resource Center for *Toxoplasma* for providing strains and the French National Reference Center for toxoplasmosis for data regarding human strains. We thank Aurélien Dumètre and Emmanuelle Gilot-Fromont for their advice on estimating the generation time of *Toxoplasma gondii*. We would also like to thank Gordon Langsley for revision of the English language. Genome sequencing was performed by the genomic platform GENOM'IC of Cochin Institute in Paris. Animal work was conducted in the animal facility managed by the technical platform BISCEm (US042 Inserm—UMS 2015 CNRS) of the University and CHU of Limoges. The computations presented in this article were carried out on the CALI calculator of the University of Limoges (CAlcul en LImousin), funded by the Limousin region, the European Union, the XLIM, IPAM, GEIST institutes and the University of Limoges.

## Author contributions

L.G., F.P. and A.M. conceived and designed the study. L.G. performed parasite cultures. F.L. and F.A. performed sequencing. L.G. performed data analysis. L.G. and A.M. acquired funding. L.G. wrote the first draft of the manuscript. L.G. and F.P. wrote and edited the manuscript. L.G., F.P., F.A., M.A.G., M.-L.D., A.H. and A.M. revised the manuscript. All authors read and approved the contents of the manuscript.

## Competing interests

The authors declare no competing interests.
