## [Peer Review File · Nature Communications]

Reviewers' comments:

Reviewer #1 (Remarks to the Author):

In this complicated but fascinating manuscript, the authors report results from whole genome sequencing of *Toxoplasma gondii* isolates from around the world, and make inferences about the demographic history of this parasite based on the synthesis of genomic and historical evidence. In general, the analyses are competently conducted and the results are clearly explained, and will be of wide interest. The Discussion is speculative, but concordant with the genomic signals observed. The impact of the manuscript could be greater if accompanied by functional validation of the presumed selected *T. gondii* mutations conferring adaptation to domestic cats, but this work will presumably follow in a subsequent manuscript. This manuscript could be improved through clarification of a few issues:

1) It would be helpful to clarify which isolates are wild vs. domestic in all figures, most particularly figure 3. Further, it would be helpful if the authors addressed the asymmetry in sampling wild vs. domestic isolates from the New World vs. the Old World in their analyses; do any potential biases arise due to lack of sampling wild isolates in the Old World? Do prior studies of wild isolates in other felid species support or refute the conclusions drawn about adaptation to domestic cats?

2) Figure 4 is very difficult to understand, with many tiny elements. What is the main point of this figure, and could this be included in the title? It makes a complicated analysis more difficult to apprehend.

Small points:

Figure 1: Colombia is misspelled

Figure 2: nearly impossible to read text-simpler to use colored symbols to represent samples at branch tips?

Figure 3-mark location of candidate selected locus? The coded sample names are less helpful than if the geographic origin of the samples could be more clearly indicated.

Reviewer #2 (Remarks to the Author):

Toxoplasma is arguably the most successful parasite in the world, as a single species can infect all warm-blooded animals and exists in a remarkable number of ecological niches. The vast majority of global clinical isolates (outside of S. America) come from one of three clonal lineages that are a familial clade, though the population is much more diverse in S. America. This asymmetrical global population structure has often been argued to have been due to a founder's effect where a small number of strains "hitched" onto e.g. rats on colonial ships and then dispersed throughout the world. Given the large population diversity in S. America compared to the rest of the world, the most parsimonious model was often considered to be that Toxoplasma originated in e.g. Amazonian Brazil. The present study provides a massive new set of sequencing data to better define global population structure. These data, combined with the highest-quality estimate to-date of the mutation rate of parasites passed continuously through mice allowed the authors to argue that the parasite disseminated from Old World to New World rather than vice versa. The authors also argue that a small genomic region is correlated with "domesticated" parasite strains, which they suggest is therefore correlated with sexual recombination in domesticated (vs. wild) cats.

Overall, this is an intriguing paper, and the data will be incredibly useful to the community. Overall the analyses appear to have been carried out carefully. However, it appears that the authors set out to prove their model for maximum splash rather than merely analyze the data and see where it led them. While the authors argue that their model is most parsimonious, the argument does not convince this reviewer (which isn't to say the discussion isn't worthwhile or that I am unwilling to be convinced). While it is always important to a field to have long-held models disrupted, paradigm-shifting work must hold itself to the highest standards.

Major comments:

- The vastly different recombination rates among the S American (frequent) vs. "domesticated" (incredibly rare; hence clonal populations) would be expected to complicate TMRCA estimates; one could not use the same mutational model for both populations. Also, the authors use Tang's equation to estimate TMRCA, but according to Tang's original publication one of the stated assumptions of Tang's analysis is that there is no recombination (fine in the clonal lineages, not so fine in the S. American population). This appears to call into question the accuracy of their estimates, upon which the majority of their arguments are based.

- The argument considered most parsimonious by the authors is that the clonal expansion of *Toxoplasma* is correlated with domestication of cats, i.e. genetic differences between domesticated vs. wild cats. But Ottoni et al (ref 40 in citation) show that very few genetic changes are correlated with cat domestication, and those that do vary are mostly in coat coloring. How could this have altered sexual recombination frequency? Also, *Felis silvestris lybica* and related species are still prevalent throughout the Mediterranean and Northern Africa, and North America is famous for large stretches of undeveloped land with many prevalent species of wild cats (for which e.g. the Canadian “cougar” strain is named from a scat-infected well), yet N America does not share the genetic diversity found in S. America – so does the argument hold that the major difference between Old World and New World *Toxoplasma* isolates being the loss of wild cat species?

- More on the last point – since divergent S. American strains have been demonstrated to be able to superinfect and recombine experimentally in domestic cats, isn't that inconsistent with the authors' model?

- Alternative hypothesis ignored by authors: *Toxoplasma* is unusual in that it does not require transmission through its definitive host to infect a new organism. Thus organisms that will eat meat, e.g. humans, hawks, rats (to some extent), can all get infected from eating an infected animal. Perhaps that plays a role in the “domestication” population.

Minor comments:

- Labels in most figures are quite small and not legible on a printed page

Reviewer #3 (Remarks to the Author):

A unique *Toxoplasma gondii* haplotype accompanied the global expansion of cats. Galal et al.

This paper reports 156 *Toxoplasma gondii* genomes, of which 105 have not previously been analyzed at WGS resolution. The authors claim that they have identified a region on Chromosome Ia, which they refer to as a unique haplotype, that is common to clonal strains of *Toxoplasma* that infect and expand in domestic cats. This finding has been extensively published on previously (please see Khan, *Genome Res*, 2006; Khan, *PNAS*, 2007; Khan, *mBio*, 2011), just not at WGS resolution. The authors argue that by combining environmental and functional data, this region (which comprises 0.16% of the *T. gondii* genome) has been selected because it promotes sexual reproduction in domestic cats. But there is no experimental validation to back up this claim. They produced a direct estimate for the *Toxoplasma gondii* mutation rate. Based on SNP variation identified within 50 Type II isolates, they suggest that the clonal Type II lineage emerged 13K to 50K years ago, just prior to the domestication of cats. They argue that domestic *T. gondii* lineages carrying the same Chromosome Ia haplotype are more efficient at sexual expansion in domestic cats, but again, they do not provide the experimental data to prove this fact. Rather, they reference one paper that assayed a single wild strain of *T. gondii* that has a defect in

self-mating as their evidence. Herein lies a fundamental problem with the author's interpretation, since the vast majority of wild strains are readily infectious in mice, produce transmissible cysts, and can be expanded as highly fecund infections in domestic cats (see Khan, PNAS, 2007; in addition to the vast collection of wild strains generated at the USDA by Dr. JP Dubey, who recovered a majority of these wild strains by assaying them through domestic cats). What is novel, is the use of Population Branch Statistics to further resolve the previously identified Chromosome Ia region to a limited set of polymorphic genes that should certainly be tested for their ability to promote sexual reproduction in domestic cats in an allele-specific manner to substantiate the author's claims. Without this, the paper fails to provide sufficient evidence to support this region as one potential explanation for the global expansion of specific *T. gondii* clonotypes by domestic cats.

Major Points:

1. The title is both misleading and insufficiently supported. The unique haplotype refers to a region on Chromosome Ia that is common to clonal lineages I, II, III. But this region is divergent in clonal lineage HG12, why was this not discussed? HG12 isolates produce highly fecund infections in domestic cats and are considered the 4th clonal lineage of North America. One study previously showed that Type II and HG12 cause essentially the same frequency of *T. gondii* infections in feral domestic cats on the West Coast of the USA (Van Wormer, 2014, PLoS NTD). These data certainly challenge their thesis that the region on Chromosome Ia common to Types I, II, III is responsible for the selective expansion of clonotypes through domestic cats. Further, the molecular clock has not been rigorously calculated, and is rife with assumption biases (see below), so it is impossible to accurately infer the timing of the molecular origin of the domestic Type II clonal lineage. Suggesting that it pre-dates the domestication and global expansion of cats is not based in fact, and borders on wishful thinking.
2. The authors identified 1,262,582 variant positions that were shared across the 156 genomes sequenced. They then performed what they called a "clone-censoring" step to remove 85 isolates (from what I gather, it was 2 Type I, 49 Type II, 17 Type III, 14 Africa 1, 3 Africa 3), which reduced the number of variant positions to only 588,777 SNPs (or by 53%). The argument made was that diversity within a lineage is minimal, so this diversity would not influence the ancestry plots that showed extant admixture only among the non-clonal domestic strains from the Old and New World (Supplementary Figure 1). However, by removing the heterogeneity present among the clonal lines, the analysis presented a strongly biased and potentially artefactual interpretation of the data. Figure 2 clearly suggests that significant diversity exists within clonal strains Type II, III and Africa 1, although no scale was provided in the figure, so it is impossible to gauge the true number of SNP differences among strains within each lineage. What happened to ~674,000 SNPs? Supplementary Figure 6 establishes that admixture among domestic Type II, Type III and Africa 4 clonal strains has occurred, which would surely affect the TMRCA calculations. If all "clone-censored" strains are included, how does this affect the "timing" of the molecular origin of the clonotypes with respect to the domestication of cats?
3. Figure 3 needs to be radically overhauled. The ancestry plots suggest that Type I, II, III and Africa 4 do not exist as admixture clones, when in fact multiple studies (Boyle PNAS, 2006; Lorenzi, NatComm, 2016; Zhang, 2017, Mol Biol Evol) clearly establish these lineages as admixtures. Hence, the single color hue across each chromosome by clonotype belies truth. For example, Type II shares chromosomes Ia and IV with Type I strains, and Type II shares chromosomes Ia, XI, and parts of Ib, III, VI, VIIb, VIII, IX and XII with

Type III. Figure 3 needs to be re-imagined, it fails to reflect truth, each clonal strain is an admixture of different colors. The cut-offs also do not coincide with the analyses performed in Figure 4 and Supplemental Table 4. For example, the HG12 strains WdUS01 and WdUS04 share the same color hue (green; Type II) on chromosome III with two recombinant strains (WdUS02, WdUS03), but the analysis presented in Supplemental Table 4 for these 4 isolates suggest that the two HG12 strains are at least 47K-177K years distant from the two recombinant strains that share the same color at this chromosome. This one example calls into question the entire dataset.

4. Molecular Clock calculation. The authors do not possess reliable data on the passage history for the strains they used to estimate natural variation. Lab strains separated by 30 years, with no reliable estimate for the true number of passages they have undergone, nor what manipulations they have been exposed to, is not reliable. If the authors want to include such data, they need to carefully record the number of passages between each line and sequence multiple clones from independent cultures across defined time points to truly achieve a reliable result, as has been done previously for *Plasmodium*. In their own words, they state (line 267; TMRCA estimates are very sensitive to sampling and obtaining accurate estimates depends on robust sampling of source populations). Too many unreliable estimates are extant in the literature, and the authors have the methodology in place to do this precisely. Furthermore, how does their result compare to other TMRCA calculations performed for *T. gondii* based on examining drift in introns across a large number of *T. gondii* strains?

5. The data estimates presented in Figure 4 suggest that the time between the two most divergent genomes of Type II is significantly greater than for the other clonotypes, which they argue are more recently derived. However, this result more likely reflects their bias in sampling. They had 48 Type II strains to draw from, whereas they only had 3 Type I strains (Supplemental Table 8). Importantly, there existed about 500 SNPs between the 3 Type I strains. Assuming the accumulation of SNPs is similar among the different clonotypes (across time), 3 goes into 48 sixteen times (or $500 \times 16 = 8000$ SNPs) which is nearly equivalent to the number of SNPs that separate the two most divergent Type II lineages. Which may suggest that Type I is as OLD as Type II and that the Figure 4 analysis is potentially inaccurate and prone to an oversampling bias.

6. The penetrance of the Chromosome Ia haplotype among successful strains of *T. gondii* that expanded to represent a majority of infections in North America and Europe is certainly interesting (and has been identified previously). While it is possible that the PBS analysis performed using the strains in this study support a region on Chromosome Ia as potentially relevant for the success of these strains, the authors failed to prove that a gene in this region is responsible for the expansion of these strains among domestic cats, as they strongly suggest. Importantly, this analysis fails to include a large number of HG12 isolates, that are likewise highly successful clones, that produce highly fecund infections in domestic cats, but do not share the same chromosome Ia haplotype. It is therefore incumbent on the authors to provide formal proof, that a gene in this region is both necessary and sufficient to promote sexual reproduction in domestic cats to rationalize the success of these clonotypes, as they report.

Minor Points:

1. Which is it? For Type II, Fig 2 lists 47 isolates, Supplemental Table 1 lists 50 isolates, Supplemental Table 8 lists 48 isolates. Please be consistent. If strains are dropped from particular analyses, a rationale should be provided.

2. Line 278 – “...TMRCA estimates indicate that massive introgressions of types I, II, II and Africa 1 into New World populations have occurred...” I think the authors are suggesting here that Old World lineages have recombined multiple times with New World strains to produce admixture lines that exist as chimeras – is that correct?

3. The authors renamed the vast majority of strains to conform with their “domestic” vs. “wild” designation and geographic origin of the isolate. This was quite a distraction, to sort out which designation was GT1, Me49, VEG, etc. The organization of the Supplemental Tables was difficult to infer. Is it possible to add the common name to Supplemental Table 2, and also re-sort Supplemental Table 2 to be by multilocus Type, then by country? Or if the preference is to sort by country, then please sort by multilocus type.

4. In the Supplementary File, line 383, the authors state Type III genomes (n=19), but in all other places within the manuscript, they state that only 18 Type III genomes were analyzed – which is it?

5. The authors propose a model in which rodents in the New World are multiply co-infected with different strains of *T. gondii* to support the increased diversity detected in isolates from the New World. Do the authors have evidence to support such a claim, or have studies been done that they can reference that indicate such a high frequency of mixed infections to support their proposed model?

**Reviewer #1 (Remarks to the Author):**

In this complicated but fascinating manuscript, the authors report results from whole genome
sequencing of *Toxoplasma gondii* isolates from around the world, and make inferences about the
demographic history of this parasite based on the synthesis of genomic and historical evidence. In
general, the analyses are competently conducted and the results are clearly explained, and will be of
wide interest. The Discussion is speculative, but concordant with the genomic signals observed. The
impact of the manuscript could be greater if accompanied by functional validation of the presumed
selected *T. gondii* mutations conferring adaptation to domestic cats, but this work will presumably
follow in a subsequent manuscript. This manuscript could be improved through clarification of a few
issues:

**AUTHORS' RESPONSE – We are grateful to the reviewer for accepting to revise our work and we**
**thank him for his positive remarks.**

1) It would be helpful to clarify which isolates are wild vs. domestic in all figures, most particularly
figure 3.

**AUTHORS' RESPONSE – We modified Fig. 3 according to the reviewer's recommendation.**

Further, it would be helpful if the authors addressed the asymmetry in sampling wild vs. domestic
isolates from the New World vs. the Old World in their analyses; do any potential biases arise due to
lack of sampling wild isolates in the Old World?

**AUTHORS' RESPONSE – We thank the reviewer for this interesting question.**

**Indeed, in the Old World, we observe that only domestic strains (harbouring the domestic haplotype**
**on chr1a) are found and are infecting both domestic and wild animals.**

**Domestic cats have invaded and proliferated in all continents (the world population is estimated at**
**600 million individuals (Baker, 2010)). However, domestic cats have existed in the Old World for**
**millennia and were only introduced in the New World in the last few centuries. Therefore, it is likely**
**that their effect on *T. gondii* populations is more profound in the Old World compared to the New**
**World. Globally, most species of wild felids are undergoing important decline, but the phenomena is**
**more massive in most regions of the Old World (source IUCN). The populations of most Old World**
**species of felids (Lions, Tigers, Cheetahs) are today ridiculously low (refer to table below). An**
**example: Lynx populations in Europe (*Lynx lynx*) do not exceed 10,000 individuals, less than 30,000 in**
**each of China and Russia, whereas Lynx (*Lynx rufus*) in the USA are about 3,000,000.**

**At first sight, providing evidence for the genetic dichotomy between domestic and wild strains in all**
**continents would have given more strength to our model. Intense sampling efforts could maybe**
**enable to identify true wild strains in the Old World and it will be for sure interesting to study their**
**genomes. In France, only type II and type III were found despite isolating the several hundred *T.***
***gondii* strains. However, in a way, the exclusive occurrence of domestic strains in all environments in**
**the Old World is consistent with our hypothesis: the environmental distribution of ~100 kb**

haplotypes (domestic *versus* wild) is not associated with the type of environment *stricto sensu*, but
rather to the species of definitive hosts (history of occurrence, sizes of population) that are
contaminating the environment with *T. gondii* oocysts.

We are clarifying these points in the new version of the manuscript.

Do prior studies of wild isolates in other felid species support or refute the conclusions drawn about
adaptation to domestic cats?

AUTHORS' RESPONSE – Of course, the use of domestic or wild felids in experiments is hampered by
the important ethical concerns associated with the use of live cats. Therefore few studies are
available but they still provide interesting insights. Jewell et al. (1972) and Miller et al. (1972)
conducted experimental infections of wild felids using a domestic *T. gondii* strain named M-7741
(DCUSA04). Our study shows that this strain harbours the domestic haplotype of *T. gondii*. Only one
out of six seronegative bobcats (*Lynx rufus*) excreted oocysts after being challenged with M-7741.
None of the two seronegative margay cats (*Leopardus wiedii*) challenged with the same strain
excreted oocysts. However, one seronegative cougar (*Puma concolor*) and one seronegative
Jaguarondi (*Puma yagouaroundi*) challenged with this strain excreted oocysts. Interestingly, two
Asian leopard cats (*Prionailurus bengalensis*) excreted oocysts after being challenged with LRH strain,
but not after a previous challenge with M-7741 strain. LRH strain was previously isolated from the
faeces of a Panamanian ocelot (*Felis pardalis*), and is therefore probably a true wild strain (Jewell et
al., 1972) but this is not sure of course. Overall, these results suggest that domestic strains are poorly
transmitted by wild felids but it is for now impossible to demonstrate clear species-specific patterns
based on these small samples.

2) Figure 4 is very difficult to understand, with many tiny elements. What is the main point of this
figure, and could this be included in the title? It makes a complicated analysis more difficult to
apprehend.

AUTHORS' RESPONSE – We enlarged the figure and modified its title and legend (now Supplementary
Fig. 6).

In organisms having frequent recombinations (humans for example), a genomic sequence can be a
succession of very small portions of sequences having different ancestries due to the accumulation of
ancestry mixing by crossovers through generations. This obviously complicates the calculations of
TMRCA, and therefore many different sophisticated models have been developed to resolve this
issue. This is not the case in our study as recombinations in the here considered *T. gondii* populations
are infrequent. Many strains have inherited "intact" chromosomes of single ancestry (Fig.3) showing
no evidence of crossovers on them. These chromosomes, although inherited following a process of
sexual reproduction, are clones of one of their respective parental strains.

Our objective was to demonstrate that (1) intercontinental lineages recently spread from the Old
World to the New World and that (2) domestic New World strains are the progeny of these Old
World lineages? To do so, we first identified all chromosomes having a single ancestry among
putative hybrids from the plots on figure 3. We used these "intact" chromosomes of single ancestry
and estimated their divergence (TMRCA) from Old World lineages. For example, if a domestic
Brazilian strain had a full ancestry of type III at chromosome 2 (blue in Fig. 3), we compared the

sequence of this Brazilian chromosome to all chromosome 2 sequences of all types III strains from
the Old World. Finding a TMRCA of ~500 years would support the two hypotheses mentioned in the
beginning of this paragraph. In the figure, the different colours represent the different ancestries and
the lengths of ellipses correspond to the interval of TMRCA (e.g. 600-300 years).

Small points:

Figure 1: Colombia is misspelled

**AUTHORS' RESPONSE – We corrected this mistake.**

Figure 2: nearly impossible to read text-simpler to use colored symbols to represent samples at
branch tips?

**AUTHORS' RESPONSE – We enlarged the font of individual labels.**

Figure 3-mark location of candidate selected locus? The coded sample names are less helpful than if
the geographic origin of the samples could be more clearly indicated.

**AUTHORS' RESPONSE – No, the coloured plots of Fig. 3 were produced from output text files of**
**Ancestry_HMM software. So the exact boundaries of the chromosomal segments of different**
**ancestries were extracted from these files. We clarify this point in the new version of the manuscript.**

**Reviewer #2 (Remarks to the Author):**

Toxoplasma is arguably the most successful parasite in the world, as a single species can infect all
warm-blooded animals and exists in a remarkable number of ecological niches. The vast majority of
global clinical isolates (outside of S. America) come from one of three clonal lineages that are a
familial clade, though the population is much more diverse in S. America. This assymetrical global
population structure has often been argued to have been due to a founder's effect where a small
number of strains "hitched" onto e.g. rats on colonial ships and then dispersed throughout the world.
Given the large population diversity in S. America compared to the rest of the world, the most
parsimonious model was often considered to be that Toxoplasma originated in e.g. Amazonian Brazil.
The present study provides a massive new set of sequencing data to better define global population
structure. These data, combined with the highest-quality estimate to-date of the mutation rate of
parasites passed continuously through mice allowed the authors to argue that the parasite
disseminated from Old World to New World rather than vice versa. The authors also argue that a
small genomic region is correlated with "domesticated" parasite strains, which they suggest is
therefore correlated with sexual recombination in domesticated (vs. wild) cats.

Overall, this is an intriguing paper, and the data will be incredibly useful to the community. Overall
the analyses appear to have been carried out carefully. However, it appears that the authors set out
to prove their model for maximum splash rather than merely analyze the data and see where it led

them. While the authors argue that their model is most parsimonious, the argument does not
convince this reviewer (which isn't to say the discussion isn't worthwhile or that I am unwilling to be
convinced). While it is always important to a field to have long-held models disrupted, paradigm-
shifting work must hold itself to the highest standards.

**AUTHORS' RESPONSE** – We thank the reviewer for the positive remarks about the interest of our
manuscript and we are grateful to him for his sincere will to help in improving the consistency and
the robustness of the conclusions drawn from our results.

We regret this impression of “maximum splash” as it was not at all our intention. In the revised
version of the manuscript, we fully restructured the discussion. We anchor it on the major findings of
our study in a more systematic way before developing our interpretation.

Before answering to each of the reviewer's specific comments, we would like to comment some of
the assertions developed in his opening paragraph.

The reviewer's argues that the most parsimonious model was often considered to be that
*Toxoplasma gondii* originated in South America and that our study is disrupting a long-held model.

First, we would like to point that the objective of our study was to focus on recent history of *T. gondii*
in relation to domestication (see I.108-109). It was not designed to answer to the question of the
origin of *T. gondii*. We do not argue that the parasite disseminated from Old World to New World,
but rather that the domestic haplotype of *T. gondii* followed this route. In addition, our results
provide some insights about the genetic proximity between type II, Chinese 1 and haplogroup 12. By
crossing our TMRCA estimates (late Pleistocene) to historical data on animal migration at this time,
we hypothesized that *T. gondii* migrations took place from Asia to North America at this time and not
the opposite. This hypothesis is specific to *T. gondii* strains of clade D (type II, Chinese 1 and
haplogroup 12). However, we do not pretend at all to provide evidence for an Old World origin of *T.*
*gondii* as a species.

Secondly, we would like to point that no consensus exists about the ancestral geographical origin of
*T. gondii*. Bertranpetit et al. (2017) IGE provided a model supporting a South American origin,
whereas other studies proposed an origin in North America Khan et al. (2007) PNAS ; Minot et al.
(2012) PNAS. *T. gondii* emerged in the wild several millions years ago. Therefore the history of its
origin is a strictly wild history and needs wild isolates to be understood. With the benefit of hindsight,
we think today that the absence of wild strains from the Old World in all above mentioned studies
(only wild strains from North and South America were available) does not enable to answer to the
question of the origin at this stage. That is why we are only addressing the question of recent history
in the present study.

Major comments:

- The vastly different recombination rates among the S American (frequent) vs. “domesticated”
(incredibly rare; hence clonal populations) would be expected to complicate TMRCA estimates; one
could not use the same mutational model for both populations.

AUTHORS' RESPONSE – Genetic evidence (from this study and from previous ones) supports various
degrees of clonality among different populations (according to geography and to ecotype). However,
we do not have precise knowledge of the recombination rate of *Toxoplasma gondii* in the literature.

With all the respect due to the reviewer, we do not agree with its distinction between “the S
American (frequent) vs. “domesticated” (incredibly rare; hence clonal populations)” for two reasons:
(1) Domestic clonal populations are common in South America (e.g. Caribbean 1,2,3) so opposing
South American strains to domesticated strains or to clonal populations is not relevant in our
opinion. (2) Our local ancestry analyses (Fig.3) clearly show that recombinations are rare in domestic
South American populations. Indeed, we do not observe a fine mosaic of different ancestries
alternating across these genomes but we rather observe large blocks (often reach the full length of
chromosomes) of single ancestry, a pattern indicative of rare crossovers. By considering the genomic
pattern of recombination obtained from a single experimental recombination between two *T. gondii*
strains (Khan et al., 2014b BMC Genomics; FIG.3), it appears that the ancestry pattern we have in our
study (Fig. 3) is consistent with the idea that few rounds of recombination were sufficient to give rise
to South American domestic populations. However, we agree that the recombination rate of these
populations appears to be greater than that of domestic populations in North America, and also that
of Old World populations, for which recombinations are "incredibly rare" events.

The question of the methodology used in TMRCA estimation is of course crucial.

We all know that crossovers occurring between two sequences during recombination result in a new
sequence which is a mosaic of the two parental sequences, and these parental sequences often have
very different histories. Now if we calculate the TMRCA separating this new sequence from other
sequences, this could not be straightforward since the TMRCA will vary according to the portion of
the new sequence that is considered (portions inherited from parent 1 or parent 2). In organisms
having frequent recombinations (humans for example), a genomic sequence can be a succession of
very small portions of sequences having different ancestries due to the accumulation of ancestry
mixing by crossovers through generations (some examples can be found in these studies Henn et al.,
2012; Fitak et al., 2018; Kim et al., 2020). This obviously complicates the calculations of TMRCA, and
therefore many different sophisticated models have been developed to resolve this issue.

This is not the case in our study as recombinations in the here considered *T. gondii* populations are
infrequent. Indeed, many strains have inherited “intact” chromosomes of single ancestry (Fig.3)
showing no evidence of crossovers on them. These chromosomes, although inherited following a
process of sexual reproduction, are clones of one of their respective parental strains. We only used
these “intact” chromosomes in our TMRCA calculations.

Also, the authors use Tang's equation to estimate TMRCA, but according to Tang's original
publication one of the stated assumptions of Tang's analysis is that there is no recombination (fine in
the clonal lineages, not so fine in the S. American population). This appears to call into question the
accuracy of their estimates, upon which the majority of their arguments are based.

AUTHORS' RESPONSE – This would have been true if we had calculated TMRCA between whole
genomes. But as we clarified above, we only used single ancestry chromosomes to calculate TMRCA,
enabling us to get rid of the bias introduced by recombination in TMRCA calculation. Therefore,
Tang's equation appears to be appropriate in our case.

- The argument considered most parsimonious by the authors is that the clonal expansion of
Toxoplasma is correlated with domestication of cats, i.e. genetic differences between domesticated
vs. wild cats. But Ottoni et al (ref 40 in citation) show that very few genetic changes are correlated
with cat domestication, and those that do vary are mostly in coat coloring. How could this have
altered sexual recombination frequency?

**AUTHORS' RESPONSE** – We rather argue that the expansion of the domestic cat caused a selective
sweep in the areas invaded by this host and enabled the dissemination of the domestic haplotype of
*T. gondii* on chromosome 1a. We are not associating the expansion of domestic cats to more
clonality, simply because our study does not show that recombination is more frequent in the wild
compared to the domestic environment. The question of differences in recombination frequency
between domestic and wild environments was not the aim of our study.

We argue that the cat's adaptation haplotype of *T. gondii* emerged in the domestic environment with
the domestication and expansion of cats. Given that today-domestic cats are still genetically very
close from their wild relatives of species *Felis silvestris* (at least subspecies *F. s. lybica* according to
Ottoni et al.), it is justified to claim that the cat's adaptation haplotype of *T. gondii* is probably not
strictly domestic and is probably efficiently disseminated by at least *F. s. lybica* and maybe other *Felis*
*silvestris* (all of them being restricted to the Old World, which is an important point to recall). Here,
we will answer by quoting a paragraph from the website of the International Union for Conservation
of Nature (IUCN) (<https://www.iucnredlist.org/species/60354712/50652361#population>): "The
world's population of domestic cats, *Felis catus*, was estimated as c. 600 million (Baker et al. 2010),
making the domesticated descendant of *Felis silvestris* one of the world's most numerous animals.
However, domestic cats hybridise readily with Wildcats, and genetic analysis of "wildcat" samples
found that most populations showed evidence of hybridisation (Nowell and Jackson 1996, Driscoll et
al. 2007). There are probably very few, if any, "wildcat" populations which have little history of
hybridisation with domestic cats." More details about population sizes of wild cats in different
countries can be found on this page (also refer to table below), and we can easily notice huge
differences between wild cat and domestic cat populations (usually few hundreds or thousands
versus millions, respectively).

Also, *Felis silvestris lybica* and related species are still prevalent throughout the Mediterranean and
Northern Africa, and North America is famous for large stretches of undeveloped land with many
prevalent species of wild cats (for which e.g. the Canadian "cougar" strain is named from a scat-
infected well), yet N America does not share the genetic diversity found in S. America – so does the
argument hold that the major difference between Old World and New World Toxoplasma isolates
being the loss of wild cat species?

**AUTHORS' RESPONSE** – The question of the reviewer here is of course a crucial element for the
consistency of the paradigm we develop in our study and we are therefore grateful to him for asking
this question. In the recent review of *T. gondii* genotypes in North America (Jiang et al., 2018 FIG.1),
we can see that many genotypes are still persisting in the wild in this continent. Of course the ~ 70
million domestic cats found in North America have huge environmental impact, with oocysts shed by
these domestic hosts reaching wildlife (even marine wildlife) by spreading over long distances via
waterways (Dabritz and Conrad, 2010; VanWormer et al., 2013a). This explains the high frequency of
domestic genotypes (type II, type III) among wildlife from this continent.

Also, important heterogeneity exists within South America when we compare wild isolates from
different countries of the continent. It is clear that the most diversified population of *T. gondii*
described to date is occurring in the Amazonian forest of French Guiana, a relatively well-preserved
environment with 8 species of wild felids cohabiting in this environment, which is a quite unique
situation (Mercier et al., 2011). The wild environment is far less preserved in many regions of South
America. In Brazil for example, strains isolated from wild animals were often also found in domestic
animals. One could argue that these environmental interpenetrations of strains could blur the signal
of the distinction between domestic and wild strains as we are presenting it in our study. What we
clearly see is that finding a strain in the domestic environment not harbouring the cat's adaptation
haplotype is exceptional and such strains fail to occupy domestic niches even in situations where
domestic and wild environment are in close contact (e.g. French Guiana).

We clarify these points in the revised manuscript.

We hope the previous answer provide enough clarification regarding the specific example of *Felis*
*silvestris lybica*.

- More on the last point – since divergent S. American strains have been demonstrated to be able to
superinfect and recombine experimentally in domestic cats, isn't that inconsistent with the authors'
model?

AUTHORS' RESPONSE – We thank the reviewer for this important question. A host infected with *T.*
*gondii* develops good immunity (not 100% though) against new infections. However, this immunity
appears less efficient when the new infection involves a highly divergent strain, allowing what is
called superinfection to happen. Here comes the importance of defining what a divergent strain is
and “divergence” is of course a relative concept. According to our results and to previous results (Su
et al., 2012; Lorenzi et al., 2016), it is clear for example that wild Amazonian strains are highly
divergent from the major domestic types (types I,II,III and Africa 1). Our results also show that wild
type 12 strains (RFLP lineage #5) from North America are “moderately” divergent from type II strains.
Importantly, domestic South American strains and wild South American strains should not be
considered one and the same, as the latter are a mixture of wild South American strains and major
domestic lineages. Domestic South American strains have inherited large portions of their genome
from types I,II,III and Africa 1 and this happened very recently according to our dating estimates.
Cross-immunity between these two groups is therefore likely to occur, but this will likely be
determined by the inheritance (or not) of alleles of genes involved in immunological recognition.
Anyway, it is important to recall that superinfected hosts were rarely found in real life. This rarity fits
well with our model, which supports rare recombinations (see Fig.3), knowing that rare
recombinations are probably the result of rare superinfections.

Experimental results suggest that wild strains are less efficiently excreted in the form of oocysts by
domestic cats compared to domestic strains (Khan et al., 2014a). Therefore, sexual reproduction
involving a wild strain in a domestic cat is probably a rare event in natural conditions. Again, this
rarity fits well with our model, which supports rare recombinations (more frequent than in other
parts of the World though).

Alternatively, if we assume that most domestic South American strains are effective at frequently
superinfecting hosts, we are breaking a major barrier for sex in the *T. gondii* cycle, and there will

likely be much less clonality in South American domestic populations, which is not the case (Pena et
al., 2008; Mercier et al., 2011).

- Alternative hypothesis ignored by authors: *Toxoplasma* is unusual in that it does not require
transmission through its definitive host to infect a new organism. Thus organisms that will eat meat,
e.g. humans, hawks, rats (to some extent), can all get infected from eating an infected animal.
Perhaps that plays a role in the “domestication” population.

**AUTHORS’ RESPONSE** – The ability of *T. gondii* to infect all warm-blooded species is indeed a
remarkable characteristic of this species. However, hosts that will truly exert a selective pressure
over *T. gondii* populations are hosts that get infected AND subsequently transmit the parasite. In the
domestic environment, cats and their prey (mainly rodents, and then birds) can do this efficiently and
are the main actors of the cycle. Other domestic hosts (e.g. human, dogs, livestock) often get
infected but seldom transmit the parasite. A paper discusses very nicely the idea of difference in host
importance and explains the notion of “evolutionarily significant host”: Müller, Urs B., and Jonathan
C. Howard. "The impact of *Toxoplasma gondii* on the mammalian genome." *Current opinion in*
*microbiology* 32 (2016).

We clarify this point in the revised manuscript.

Minor comments:

- Labels in most figures are quite small and not legible on a printed page

**AUTHORS’ RESPONSE** – We enlarged the font of individual labels in most figures when relevant.

**Reviewer #3 (Remarks to the Author):**

A unique *Toxoplasma gondii* haplotype accompanied the global expansion of cats. Galal et al.

This paper reports 156 *Toxoplasma gondii* genomes, of which 105 have not previously been analyzed
at WGS resolution.

The authors claim that they have identified a region on Chromosome 1a, which they refer to as a
unique haplotype, that is common to clonal strains of *Toxoplasma* that infect and expand in domestic
cats. This finding has been extensively published on previously (please see Khan, *Genome Res*, 2006;
Khan, *PNAS*, 2007; Khan, *mBio*, 2011), just not at WGS resolution.

**AUTHORS’ RESPONSE** – We are grateful to the reviewer for accepting to revise our work.

The reviewer questions the novelty of our results compared to what has been published previously.

We are mentioning the contribution of Khan et al. in our manuscript l.359-360: “Note that a number
of strains shared the same haplotype for the whole length of chromosome 1a, a pattern previously
noticed in past studies” and we are citing the main paper on this topic (Khan et al., 2007 *PNAS*).

Khan et al. noticed that many strains shared what they named “a monomorphic version of
chromosome 1a” (same allele for the whole length of chromosome 1a using several short markers).
They proposed a number of hypotheses (among them the hypothesis of adaptation to domestic
environment) to explain this pattern. However, they could not show that this “monomorphic version
of chromosome 1a” is a marker of adaptation to the domestic environment for the very simple
reason that although many domestic strains harboured this monomorphic version of the whole
chromosome 1a, many also did not (Khan et al., 2007 PNAS; Khan et al., 2011 mbio). If this
“monomorphic version of chromosome 1a” is an adaptation to the domestic environment, why many
domestic strains do not carry this allele? Khan et al. sequenced different short portions of
chromosome 1a. However, none of their markers indicate a global dichotomy between wild and
domestic strains. Since they did not use WGS as pointed by the reviewer, Khan et al. simply “missed”
the portion of chromosome 1a that distinguish domestic strains from wild ones. The Fig.S3 in Khan et
al., 2011 mbio is edifying in this regard. Therefore, their results do not in any way support a global
dichotomy between wild and domestic strains and they did not claim to have made such a discovery
in any of their studies.

Here, the use for the first time of selection inference tools combined with WGS reveals the key
pattern about chromosome 1a: we show that the genomic region marking the global adaptation to
the domestic environment is not the whole chromosome 1a (1.8 million bases), but it is rather a
100,000 bases portion of this chromosome. We show that it is precisely the consideration of this
~100 kb portion of chromosome 1a that reveals a global dichotomy between wild and domestic
strains and enables for the first time to establish a global association between this haplotype and the
domestic environment. This pattern is striking when we consider the figure 5 d (now figure 4 d) of
our manuscript. Our study shows that this pattern can be explained by the much stronger linkage
disequilibrium (often reaching the whole length of chromosome 1a) observed in domestic strains
relative to wild ones around the ~100 kb outlier region, which probably persists due to the combined
effects of selection and rarity of sexual recombinations in domestic *T. gondii* populations. Since the
conserved domestic haplotype is only a portion of chromosome 1a, we show that the appellation
“monomorphic version of chromosome 1a” used in several publications is inaccurate and misleading.

Having regard to these elements, we can assert that we have shown for the first time that we have
identified a “unique global haplotype common to almost all domestic strains worldwide and that is
under strong positive selection in the domestic environment”. We believe that this is a remarkable
finding. Moreover, by accurately defining the genomic boundaries of the global domestic haplotype,
we are able for the first time to propose a limited number of candidate genes (based on several
criteria) that could be involved in adaptation of *T. gondii* to the domestic environment. These results
constitute valuable data for future experimental studies.

Beside findings related to chromosomes 1a, we would also like to highlight other major advances
allowed by our study in the topic: For the first time, we associate the expansion of domestic cats to
the expansion of the global domestic haplotype we have identified in space and time. We show that
domestic New World populations of *T. gondii* are the result of very recent recombinations between a
number of intercontinental lineages and wild New World populations. These findings are completely
novel. We think that the idea that the emergence of domestic populations in the New World
coincides in time with the introduction of domestic cats in North and South America by European
sailors is a major finding of this study. From a wider perspective, our study illustrates nicely how two

major “events” in human history have radically changed the global landscape of a pathogen’s
diversity: (1) domestication and (2) globalisation of exchanges in the last few centuries.

There is no doubt that Khan et al. have the merit to be the first to publish studies on *T. gondii*
evolutionary history. However, more than one decade later, we believe that it is the time to revisit
this question, for several reasons:

Khan et al. were limited by major sampling gaps. For example, they propose a New World origin for
the “monomorphic version of chromosome 1a” based on less than 50 strains, with only six Old World
strains (no strains from Asia or Africa)(Khan et al., 2007 PNAS; Khan et al., 2011 mbio).

Khan et al. used the mutation rate of *Plasmodium* in their analyses of population genetics and
TMRCA (Khan et al., 2007 PNAS; Khan et al., 2011 mbio) given the lack of an estimate of *T. gondii*
mutation rate.

Khan et al. only used multilocus sequencing and we have shown with the example of chromosome 1a
that this tool fails to identify key evolutionary patterns.

We are filling these gaps in this study by analysing the largest dataset of *T. gondii* genomes produced
to date (n=156) and by providing for the first time a direct estimate of *T. gondii* mutation rate and an
estimate of its generation time. We therefore reconstruct the recent history of global dissemination
of the domestic haplotype of *T. gondii* based on the basis of much more solid data (whole genomes)
and sampling.

The authors argue that by combining environmental and functional data, this region (which
comprises 0.16% of the *T. gondii* genome) has been selected because it promotes sexual
reproduction in domestic cats. But there is no experimental validation to back up this claim.

**AUTHORS’ RESPONSE – We partially agree with the reviewer on this point.**

Full evidence to prove the selection of a trait should be supported by genomic patterns, geographic
distributions, and functional characterization. We will discuss each of these three aspects to show
how our results and previous results support our point.

Genomic evidence: Genomic evidence is supported by the results provided in Figure 5 (Population
Branch Statistics, high degree of conservation of the domestic haplotype, extended linkage
disequilibrium around domestic SNPs).

Geographic evidence: Geographic evidence is supported by the tight association between the
domestic haplotype of *T. gondii* and the environment of domestic cats (and its evolution in time,
mainly with the coincidence in the estimates between the introduction of domestic cats in the New
World and the emergence of domestic New World strains harbouring the domestic haplotype of *T.*
*gondii*).

In the domestic environment, other hosts can get infected but few play a role in transmission, as *T.*
*gondii* cycle is essentially based on transmission between cats and their prey (mainly rodents, then
birds). A host must be efficiently transmitting a parasite to be able to exert a selective pressure on
this parasite. A paper (Müller, Urs B., and Jonathan C. Howard. "The impact of *Toxoplasma gondii* on
the mammalian genome." *Current opinion in microbiology* 32 (2016)) discusses very nicely the idea

of difference in host importance and explains the notion of “evolutionarily significant host”. In this
regard, the domestic cat have a tremendous advantage over any other host given its ability to
excrete dozen of millions of highly resistant oocysts that survive during months in the environment.
House mice and rats are important hosts to consider in this frame as they are important prey of cats
and have also recently spread to many parts of the world (the New World and West Africa). It has
been suggested that house mice could play a role in selecting certain *T. gondii* strains at the expense
of others. However, selection exerted on *T. gondii* by mice was associated to the virulence of *T.*
*gondii* strains, which is not explained by the opposition between wild and domestic environments
since many mouse-virulent strains are found in both environments(Howe and Sibley, 1995; Pena et
al., 2008; Mercier et al., 2010, 2011; Shwab et al., 2018). Moreover, house mice and rats are not
found in certain areas where the domestic haplotype of *T. gondii* is well established so the
geographical association does not stand for these species (Dalecky et al., 2015; Hima et al., 2019).

Functional evidence: Domestic cats are unique in the domestic environment in that they transmit *T.*
*gondii* through the canal of sexual reproduction (cats are not eaten by other cats). Selection needs
transmission to produce its effect. Therefore, a *T. gondii* strain not adapted to be SEXUALLY
transmitted by domestic cats will not be transmitted. The question now is whether domestic cats
equally transmit all *T. gondii* strains or only strains harbouring the domestic haplotype. An
experimental study has shown that domestic strains are more efficiently transmitted by domestic
cats than wild strains (Khan et al., 2014a). The limitation of this study is that tested strains only came
from French Guiana. Our study validates *a posteriori* that the tested domestic strains (common
Caribbean lineages) harbour the domestic haplotype whereas wild strains do not. What should be
done at this stage is expanding these experiments to other domestic and wild strains from other
parts of the world. Of course performing these experiments on live cats will not be possible for
evident ethical issues (see : [https://www.science.org/content/article/scientists-decry-usdas-decision-](https://www.science.org/content/article/scientists-decry-usdas-decision-end-cat-parasite-research)
[end-cat-parasite-research](https://www.science.org/content/article/scientists-decry-usdas-decision-end-cat-parasite-research)). That is why we propose in our manuscript an alternative experimental
approach (to be optimized) instead of animal experimentation. Our study provides other important
advances in the functional characterization of the haplotype under selection: by accurately defining
the boundaries of the genomic region under selection, we are able for the first time to propose a
limited number of candidate genes. In addition, two recent studies (Ramakrishnan et al., 2019;
Farhat et al., 2020) have provided accurate data regarding the stage of expression of these genes.
This important contribution enabled to determine which genes on the ~100 kb haplotype are
expressed during sexual reproduction of *T. gondii* in cats. By crossing these findings to our results, we
were able to shed light on the few most promising genes within this haplotype to explain this
function.

We are clarifying these elements in the revised manuscript.

Finally, we think it is important to recall the fact that this is a study of population genomics and not a
mechanistic study, although it provides valuable data for future purely mechanistic studies.

They produced a direct estimate for the *Toxoplasma gondii* mutation rate. Based on SNP variation
identified within 50 Type II isolates, they suggest that the clonal Type II lineage emerged 13K to 50K
429 years ago, just prior to the domestication of cats. They argue that domestic *T. gondii* lineages
carrying the same Chromosome Ia haplotype are more efficient at sexual expansion in domestic cats,
but again, they do not provide the experimental data to prove this fact.

AUTHORS' RESPONSE – In order to avoid lengthy repetitions, we invite the reviewer to refer to our
response to its second comment, in which we have already addressed precisely this concern.

Rather, they reference one paper that assayed a single wild strain of *T. gondii* that has a defect in
self-mating as their evidence.

AUTHORS' RESPONSE – With all the respect due to the reviewer, this is not the case. This study
performed experiments on domestic cats by testing four domestic strains *versus* three wild strains
(Khan et al., 2014a).

Herein lies a fundamental problem with the author's interpretation, since the vast majority of wild
strains are readily infectious in mice, produce transmissible cysts, and can be expanded as highly
fecund infections in domestic cats (see Khan, PNAS, 2007; in addition to the vast collection of wild
strains generated at the USDA by Dr. JP Dubey, who recovered a majority of these wild strains by
assaying them through domestic cats).

AUTHORS' RESPONSE – This is an important point.

In the beginning, we would like to recall that the hypothesis of adaptation of domestic *T. gondii*
strains to sexual transmission by domestic cats have been proposed for the first time by Dr. JP Dubey
himself (Khan et al., 2014a).

The reviewer argues that “the vast majority of wild strains are readily infectious in mice, produce
transmissible cysts”. We totally agree with this statement but it is just not relevant in our case as we
are questioning in our study the different abilities of *T. gondii* strains to be transmitted by domestic
cats in the form of oocysts.

The reviewer argues that “the vast majority of wild strains [...] can be expanded as highly fecund
infections in domestic cats” and is citing Khan, PNAS, 2007. But we did not find any element to
support this statement in the study he is mentioning which did not address this topic.

The reviewer evokes “a vast collection of wild strains” belonging to Dr. Dubey. We are not aware of
this vast collection but we know from the articles published by Dr. Dubey that wild strains may not
be transmitted by domestic cats and even more: domestic strains may not be transmitted by
domestic cats (Dubey et al., 2008, 2010; Khan et al., 2014a). So this question should be addressed as
a question of differences in transmission efficiency and this what the study by Khan et al., (2014a) is
showing. To our knowledge, this is the only study that formally tested the hypothesis of different
abilities of *T. gondii* strains to be transmitted by domestic cats is the one we are citing in our study.
This study supports that domestic strains are more efficiently transmitted by domestic cats than wild
strains. In this study, we have a good idea of the dose of parasites fed to cats and the number of
excreted oocysts, two parameters that should be taken into account if we aim to compare the
efficiency of transmission between two strains. Of course, the limitation of this study is that tested
strains only came from French Guiana and we have addressed this point in the answer to the
reviewer's second comment. The studies referred to by the reviewer are not experimental studies, as
domestic cats have only been used for the "production" of oocysts, often after mouse bioassay. We
have no idea about the rationale behind the choice of strains fed to cats in each of these studies and
no idea about the dose of parasites fed to these cats nor the number of excreted.

Another crucial point is the definition of “wild strains”. Wild strains can be defined *a priori* as strains
isolated from wild animals or from humans in contact with wildlife and genetically distinct from
domestic *T. gondii* populations. Therefore, this definition supposes a very accurate knowledge of
domestic strains circulating in the region of interest and this is not the case for all North America (see
Jiang et al., 2018). In North America, our results show that strains harbouring the domestic haplotype
-beside being well-established in the domestic environment- are also reaching wildlife through
waterways and infecting wild animals (VanWormer et al., 2013b; Jiang et al., 2018). This can be
explained by the huge populations of domestic cats found today compared to populations of wild
felids that are in constant decline. This point is crucial as it implies that strains isolated from wild
animals may harbour the domestic haplotype. However, the reverse is not true, as strains not
harbouring the ~100 kb domestic haplotype are efficiently counter-selected in the domestic
environment as they are simply not found in this environment. Before asserting that a strain is a wild
strain, one should verify with genomic analysis that it does not have hybridized with a domestic
strain. Our genomic analyses show that both cases exist in the wild environment (pure wild strains
and hybrids such as WdUSA03 [B41]).

We are providing several references supporting this idea in the revised manuscript.

What is novel, is the use of Population Branch Statistics to further resolve the previously identified
Chromosome Ia region to a limited set of polymorphic genes that should certainly be tested for their
ability to promote sexual reproduction in domestic cats in an allele-specific manner to substantiate
the author’s claims. Without this, the paper fails to provide sufficient evidence to support this region
as one potential explanation for the global expansion of specific *T. gondii* clonotypes by domestic
cats.

**AUTHORS’ RESPONSE** – In order to avoid lengthy repetitions, we invite the reviewer to refer to our
response to its second comment, in which we have already addressed precisely this concern.

Major Points:

1. The title is both misleading and insufficiently supported. The unique haplotype refers to a region
on Chromosome Ia that is common to clonal lineages I, II, III.

**AUTHORS’ RESPONSE** – By only considering the Figure 5d (now Figure 4d), we can clearly see that the
unique haplotype is common to all domestic strains (except one) so we cannot agree with this
assertion. Almost all domestic strains (grey labels) cluster together within the NJ tree not only types
I,II and III .

But this region is divergent in clonal lineage HG12, why was this not discussed? HG12 isolates
produce highly fecund infections in domestic cats and are considered the 4th clonal lineage of North
America.

**AUTHORS’ RESPONSE** – We are grateful to the reviewer for bringing this point to our attention and
this is of course a very important question.

HG12 (as its name suggests) is not a clonal lineage but a haplogroup, composed of at least two major
clonal lineages (ToxoDB #4 and #5) and a number of minor lineages, all found in North America. So it
is a mistake to consider this haplogroup as a single homogenous entity (and we did this mistake). This

is important because the different lineages composing HG12 do not share the same allele for the
haplotype on chromosome 1a.

We re-examined the specific case of HG12 and what we found was very interesting and nicely
illustrates the association between the domestic *T. gondii* haplotype and the domestic environment.

The initial studies identified HG12 mainly among wild animals and therefore HG12 was first
considered as a wild haplogroup (Khan et al., 2011a). However, subsequent studies revealed that
HG12 was also common in the domestic environment. Jiang et al. (2018) published a very good
review of the genotypic diversity in North America which accurately reveals how the different
genotypes of *T. gondii* are distributed in the environment. The remarkable pattern we noticed about
HG12 in this review is that ToxoDB#4 is common in the domestic environment and rare in the wild,
whereas ToxoDB#5 shows the exactly opposite pattern. Our genomic analyses showed that
ToxoDB#4 (WdUSA02) harbours the domestic haplotype whereas ToxoDB#5 (WdUSA01 and
WdUSA04) harbours a wild haplotype. Although, we only have 3 strains representing these two
lineages, our results indicate that previous multilocus classifications (RFLP or MS) correlate strongly
(95%) with genomic classification.

Thus the pattern we observe for HG12 is one of the strongest supports to our hypothesis. This
example is very interesting in that it shows how two related lineages (#4 and #5) belonging to the
same haplogroup and found in the same region of the world segregate in space according to the type
of environment (domestic *versus* wild) and this segregation matches nicely with the different
haplotypes they harbour on chromosome 1a.

We have clarified this point in the new version of the manuscript. We renamed WdUSA02 (now
DcUSA11) to fit with this environmental evidence.

One study previously showed that Type II and HG12 cause essentially the same frequency of *T. gondii*
infections in feral domestic cats on the West Coast of the USA (Van Wormer, 2014, PLoS NTD). These
data certainly challenge their thesis that the region on Chromosome 1a common to Types I, II, III is
responsible for the selective expansion of clonotypes through domestic cats.

**AUTHORS' RESPONSE** – The authors of this study published a number of studies conducted in a very
interesting environment in which we observe an overlap (this is the term used by the authors)
between the home ranges of domestic cats and wild felids in this area. This environment is
contaminated with oocysts shed by both domestic cats and wild felids and therefore these two
groups of species can get infected with domestic and wild strains and this is what this study is
showing. However, the authors do not address the crucial question of whether or not domestic cats
infected in this area with wild strains excrete oocysts following their infection. The same authors are
asking this question in the discussion of their most recent paper on the subject (Shapiro et al., 2019
*Proceedings of the Royal Society B*), we quote “further studies on *T. gondii* oocyst genotypes shed by
domestic and wild felids would provide additional insight on sources of sea otter infection. While
Type X infections occur in both domestic and wild felids in watersheds bordering the sea otter range,
genotype data are needed for the oocysts shed by these felids. In experimental studies, the
prevalence of oocyst shedding varied with *T. gondii* strain. Greater levels of shedding were observed
in wild felids exposed to atypical ‘wild’ strains and in domestic cats exposed to archetypal ‘domestic’
strains (e.g. Types I, II or III) [39,40], but only limited genotypes were tested. One of six domestic cats

experimentally infected with an atypical strain shed similar numbers of oocysts (2×10^8) as cats
infected with domestic strains [40]. To our knowledge, shedding of Type X oocysts by a domestic cat
has only been reported for one clinically ill animal [41]. Field studies are therefore needed to clarify
levels of shedding by domestic cats infected with Type X under natural conditions.” Note: type X
refers to HG12. Therefore we can conclude that all elements brought by these studies support our
model.

Further, the molecular clock has not been rigorously calculated, and is rife with assumption biases
(see below), so it is impossible to accurately infer the timing of the molecular origin of the domestic
Type II clonal lineage. Suggesting that it pre-dates the domestication and global expansion of cats is
not based in fact, and borders on wishful thinking.

**AUTHORS’ RESPONSE** – As the reviewer is developing his concerns about our estimates of TMRCA in
comment number 4, we will provide a detailed response on this point in our answer to comment
number 4.

2. The authors identified 1,262,582 variant positions that were shared across the 156 genomes
sequenced. They then performed what they called a “clone-censoring” step to remove 85 isolates
(from what I gather, it was 2 Type I, 49 Type II, 17 Type III, 14 Africa 1, 3 Africa 3), which reduced the
number of variant positions to only 588,777 SNPs (or by 53%). The argument made was that diversity
within a lineage is minimal, so this diversity would not influence the ancestry plots that showed
extant admixture only among the non-clonal domestic strains from the Old and New World
(Supplementary Figure 1). However, by removing the heterogeneity present among the clonal lines,
the analysis presented a strongly biased and potentially artefactual interpretation of the data.

**AUTHORS’ RESPONSE** – Here we have followed basic recommendations to minimize bias related to
clonality for ADMIXTURE, a software developed for sexual organisms. If we include all strains (even
strains of the same clonal lineages), this will introduce huge bias in the analysis by deviating widely
from the assumption of sexual reproduction. The developers of this software also recommend
performing pruning of data for linkage disequilibrium (see:
<https://vcru.wisc.edu/simonlab/bioinformatics/programs/admixture/admixture-manual.pdf>).
According to the developers (see 2.4.), it is not uncommon that only few dozens of thousands SNPs
are retained after this step and this relatively small number of SNPs still provides enough resolution.
One should care on minimizing bias to fit as much as possible with the assumptions of the software’s
model much more than the number of SNPs.

Figure 2 clearly suggests that significant diversity exists within clonal strains Type II, III and Africa 1,
although no scale was provided in the figure, so it is impossible to gauge the true number of SNP
differences among strains within each lineage.

**AUTHORS’ RESPONSE** – This information is provided in Supplementary Table 8. We have revised the
NJ trees by adding the scale.

What happened to ~674,000 SNPs?

**AUTHORS’ RESPONSE** – As we pointed above, these SNPs were eliminated in certain analyses to
remove bias related to clonality when relevant.

Supplementary Figure 6 establishes that admixture among domestic Type II, Type III and Africa 4
clonal strains has occurred, which would surely affect the TMRCA calculations.

**AUTHORS' RESPONSE** – We can see on some positions with numerous SNPs on chromosome 10 of
type II strains, but this is not due to recombination as we observe no recombination breakpoints.

**Chromosome 10 of type II strains:**

For Africa 4 and type III, we invite the reviewer to reread the section “identifying clonal lineages” in
the supplementary information. As we have clarified in this section, “We therefore excluded the
divergent genomes from their respective poppr-defined lineages —which are likely the products of a
recombination with a strain of a distinct population— and generated new SNPs density plots .The
sharp variations in SNPs densities previously observed did not recur in the new plots (data not
shown), indicating that the excluded genomes had divergent ancestry in certain chromosomal
portions.”

For type III strains we clearly noticed the recombination breakpoints before removing DcUSA04.

**Type III strains before removing DcUSA04:**

**Type III strains after removing DcUSA04:**

**Of course, we can add them to the manuscript if the reviewer deems it relevant.**

If all “clone-censored” strains are included, how does this affect the “timing” of the molecular origin
 of the clonotypes with respect to the domestication of cats?

AUTHORS' RESPONSE – All strains are included (see I.136-138) for dating purposes and we did not
use the clone-censored dataset for this analysis.

3. Figure 3 needs to be radically overhauled. The ancestry plots suggest that Type I, II, III and Africa 4
do not exist as admixture clones, when in fact multiple studies (Boyle PNAS, 2006; Lorenzi,
NatComm, 2016; Zhang, 2017, Mol Biol Evol) clearly establish these lineages as admixtures. Hence,
the single color hue across each chromosome by clonotype belies truth. For example, Type II shares
chromosomes Ia and IV with Type I strains, and Type II shares chromosomes Ia, XI, and parts of Ib, III,
VI, VIIb, VIII, IX and XII with Type III. Figure 3 needs to be re-imagined, it fails to reflect truth, each
clonal strain is an admixture of different colors.

AUTHORS' RESPONSE – We understand the reviewer's concern.

In figure 3, we are defining the parental populations based on the results of global ancestry analyses
(mainly chromopainter). This analysis (Supplementary Figure 2) showed that most domestic New
World strains are sharing recent ancestry with the major intercontinental lineages on one hand and
the wild New World populations on the other hand. Therefore, Figure 3 succeeds in illustrating the
pattern of mixed ancestry of domestic New World strains which appear to be a mix between the
major intercontinental lineages and the wild New World populations. This is the key finding obtained
from this analysis.

We are aware that the major intercontinental lineages (types I, II, III and Africa 1) share common
ancestry for certain genomic loci and we are citing Boyle et al. in this sense (SI I.543).

When we aim to infer the parental populations of a hybrid population, we have to include in the
analysis a number of populations as putative parents of this hybrid population. However, in natural
populations of most species, having parental populations that are not related to each other by any
mean is very rare, so this is a "classical" concern with this kind of analyses. In the specific case of *T.*
*gondii*, the hybridizations that are involving the major intercontinental lineages are very recent and
too little time has elapsed for divergence to occur (we have produced NJ trees for each chromosome
and these trees show this pattern of absence of divergence). So it is likely that in certain genomic
positions, the software fails to distinguish between for example type II and type I or type II or type III.
However, this does not call into question the pattern of admixture that corresponds to the recent
history of encounter between to very divergent groups of populations (major intercontinental
lineages *versus* wild New World populations).

Another way to present the results could be to define only two ancestral groups: the major
intercontinental lineages one the one hand and the wild New World populations on the other hand
We could reset this analysis based on this division although we think this approach will hide many
interesting patterns.

The cut-offs also do not coincide with the analyses performed in Figure 4 and Supplemental Table 4.
For example, the HG12 strains WdUS01 and WdUS04 share the same color hue (green; Type II) on
chromosome III with two recombinant strains (WdUS02, WdUS03), but the analysis presented in
Supplemental Table 4 for these 4 isolates suggest that the two HG12 strains are at least 47K-177K
653 years distant from the two recombinant strains that share the same color at this chromosome. This
one example calls into question the entire dataset.

AUTHORS' RESPONSE –We used Ancestry_HMM for our local ancestry analyses. To our knowledge,
this is probably the only software that does not require genotypes from reference panels and that is
generalized to arbitrary ploidy, and is hence suitable for non-model haploid organisms such as *T.*
*gondii*. The software is based on a Hidden Markov Model and identifies at each position the most
probable ancestor for a hybrid strain among input ancestral populations and assign a probability to
this result. Who is the ancestor and who is the progeny depends on the assumptions obtained from
global ancestry analyses. The three (not two) HG12 strains to which the reviewer is referring were
assumed to be recombinant of a type II strain and another strain based on the result of global
ancestry analysis (mainly chromopainter). Ancestry_HMM found that type II is the most related
putative ancestral population to HG12 relatively to the other putative ancestral populations (this can
be expected as they belong to the same clade (Lorenzi et al., 2016)). However, TMRCA estimates
revealed that that the divergence between type II and HG12 probably occurred well before the
emergence of type II. Thus TMRCA estimates are very important to confirm the assumptions
obtained from ancestry analyses. For the specific case of HG12 (RFLP lineage #5) our assumptions
based on global ancestry analysis were found to be wrong and were corrected based on TMRCA
results (we clarify this point I.275).

4. Molecular Clock calculation. The authors do not possess reliable data on the passage history for
the strains they used to estimate natural variation. Lab strains separated by 30 years, with no reliable
estimate for the true number of passages they have undergone, nor what manipulations they have
been exposed to, is not reliable. If the authors want to include such data, they need to carefully
record the number of passages between each line and sequence multiple clones from independent
cultures across defined time points to truly achieve a reliable result, as has been done previously for
Plasmodium.

AUTHORS' RESPONSE – One of the most important and fascinating result we show in our study is that
domestic populations found today in the New World have emerged in the last few centuries and
their emergence coincide with the introduction of domestic cats in the Americas. Inaccurate TMRCA
estimates could of course calls into question the entire model we develop here.

We have precise knowledge of the number of passages to which laboratory strains used in the
estimation of mutation rate and separated by 30 years (RH and PRU) have been subjected and we
provide this data in details in Supplementary Methods. Three passages in mice were carried out each
686 week for RH strain during 30 years and 3 months. . Seventy-five passages in mice were carried out for
PRU strain during 30 years.

If the reviewer is referring to environmental strains, we will answer to this comment in two very
simple points:

1- All strains from this study (except four strains: DcTURKEY01 (Ankara LS1), DcUSA03(GT1),
DcUSA05(ME49), DcUSA04(M7741)) were sequenced less than 30 years after their isolation.
Isolated *T. gondii* strains are usually cryopreserved most of the time but the concern raised
by the reviewer is legitimate: we do not provide data about the number of passages
performed on these strains and new SNPs could have arisen during these passages. These
new SNPs could introduce some degree of bias in our calculations. Previously, we did not

know the mutation rate of *T. gondii* and therefore could not estimate to what extent strain
passages in the laboratory could introduce a bias in the TMRCA estimates. Fortunately, our
study provides for the first time an answer to this concern: we have estimated how many
SNPs can arise over a period of 30 years for a fast growing *T. gondii* strain subjected to an
intense regime of regular passages. We report in our study the results from an RH strain that
has been subjected to continuous in vivo culture during this long period. It reveals that 22
new SNPs arose during this period on intergenic portions of the genome. Our estimates show
that 78 – 249 years are needed for 22 new SNPs to arise in natural populations. Therefore if
we assume that some strains from our dataset have been continuously cultured during 30
705 years (which is unlikely given that this is a very heavy and unnecessary work), the maximal
bias we could have should not exceed 78 – 249 years. At the time scale of our study, this
degree of bias does not call into question our conclusions.

2- Moreover, new SNPs increase genetic distances between strains and can only lead to an
increase in TMRCA estimates (not a decrease). Therefore, if we assume a slight inaccuracy in
some of our TMRCA estimates, the exact TMRCA estimates must correspond to earlier
events. In consequence, we have another strong argument to support that this possible bias
of 78 – 249 years cannot call into question our major finding: domestic populations found
today in the New World have emerged in the last few centuries (and not earlier) and their
emergence coincide with the introduction of domestic cats in the Americas.

In their own words, they state (line 267; TMRCA estimates are very sensitive to sampling and
obtaining accurate estimates depends on robust sampling of source populations). Too many
unreliable estimates are extant in the literature, and the authors have the methodology in place to
do this precisely.

**AUTHORS' RESPONSE** – This sentence (l.267) refers to sampling not to the number of passages in the
laboratory. We have generated the largest dataset of whole genomes produced to date and we
believe that the great difficulties in obtaining *T. gondii* isolates are not unknown by the reviewer (in
comparison to *Plasmodium* for example).

Furthermore, how does their result compare to other TMRCA calculations performed for *T. gondii*
based on examining drift in introns across a large number of *T. gondii* strains?

**AUTHORS' RESPONSE** – Is the reviewer referring to the study by Khan et al. (2011) mbio?

This study conducted TMRCA calculations based on the mutation rate of *Plasmodium*. *Toxoplasma*
and *Plasmodium* have very different biology and evolutionary patterns. Now that we have the first
estimate of *T. gondii* mutation rate, we can consider that there is no reason to use the mutation rate
of *Plasmodium* in the study of *T. gondii*. We estimated the mutation rate of *T. gondii* to range
between 3.1×10^{-9} to 11.7×10^{-9} mutations per site per year, which is higher than the 1.7×10^{-9} to 3.8×10^{-9}
estimated for *Plasmodium*.

5. The data estimates presented in Figure 4 suggest that the time between the two most divergent
genomes of Type II is significantly greater than for the other clonotypes, which they argue are more
recently derived. However, this result more likely reflects their bias in sampling. They had 48 Type II
strains to draw from, whereas they only had 3 Type I strains (Supplemental Table 8). Importantly,
there existed about 500 SNPs between the 3 Type I strains. Assuming the accumulation of SNPs is

similar among the different clonotypes (across time), 3 goes into 48 sixteen times (or $500 \times 16 = 8000$
SNPs) which is nearly equivalent to the number of SNPs that separate the two most divergent Type II
lineages. Which may suggest that Type I is as OLD as Type II and that the Figure 4 analysis is
potentially inaccurate and prone to an oversampling bias.

**AUTHORS' RESPONSE** – We totally agree with the reviewer on this point and we have already
clarified it in our manuscript I.265-268 (now I.440-442): “Note that inclusion of additional isolates
from the same or different geographic areas could alter these estimates by revealing more ancient
divergence times between strains of each respective lineage. This is particularly true for type I for
which only three samples were available.” We have transferred these two sentences to the
discussion.

However, we would like to clarify a crucial point:

Lack of samples can only lead to an underestimation of the time of emergence of a given lineage (not
an overestimation). Therefore, our results provide strong support of an emergence of type II lineage
before domestication (12,980-48,988 years) and an emergence of type I, type III and Africa 1 before
the introduction of the domestic cat in the Americas. This is why we draw this conclusion I.437-440:
“Importantly, these domestic lineages emerged before dissemination of domestic cats to the New
World 500 years ago; it is hence likely that they emerged in the Old World.” This is a crucial finding
for the consistency of our model which supports that Old World lineages “accompanied” domestic
cats in their expansion from the Old World to the New World.

One could always argue that having more samples will make the estimates more accurate but this a
classical concern that can only be corrected by isolating more and more samples. The most recent
example came from the study of our own species (Hublin et al., 2017 Nature): it was revealed in 2017
that we modern humans emerged at least (so maybe earlier we do not know) 100,000 years earlier
than previously believed.

6. The penetrance of the Chromosome Ia haplotype among successful strains of *T. gondii* that
expanded to represent a majority of infections in North America and Europe is certainly interesting
(and has been identified previously). While it is possible that the PBS analysis performed using the
strains in this study support a region on Chromosome Ia as potentially relevant for the success of
these strains, the authors failed to prove that a gene in this region is responsible for the expansion of
these strains among domestic cats, as they strongly suggest.

**AUTHORS' RESPONSE** – In order to avoid lengthy repetitions, we invite the reviewer to refer to our
response to its second comment, in which we have already addressed precisely this concern.

Importantly, this analysis fails to include a large number of HG12 isolates, that are likewise highly
successful clones, that produce highly fecund infections in domestic cats, but do not share the same
chromosome Ia haplotype.

**AUTHORS' RESPONSE** – We hope our answer to the major comment number 1 of reviewer 3 provides
enough clarification regarding the specific case of HG12.

It is therefore incumbent on the authors to provide formal proof, that a gene in this region is both
necessary and sufficient to promote sexual reproduction in domestic cats to rationalize the success
of these clonotypes, as they report.

**AUTHORS' RESPONSE** – Evidence provided by this study is mainly genomic and
geographic/environmental. In addition, we provide for the first time a list of promising candidate
genes, enabling an important advance in the route of full evidence. We recall the fact that this is a
study of population genomics and not a mechanistic study, although it provides valuable data for
future purely mechanistic studies.

Minor Points:

1. Which is it? For Type II, Fig 2 lists 47 isolates, Supplemental Table 1 lists 50 isolates, Supplemental
Table 8 lists 48 isolates. Please be consistent. If strains are dropped from particular analyses, a
rationale should be provided.

**AUTHORS' RESPONSE** – There is no mistake here and we justify the exclusion of these two strains.
The first one is DcFRANCE18 and was excluded due to poor sequencing depth (1.550) and the second
one (DcMARTINIQUE01) corresponds to a strain assumed to belong to type II lineage based on MS
markers, and that was found to be a recombinant strain with WGS (see the “Identifying clonal
lineages” section of SI).

2. Line 278 – “...TMRCA estimates indicate that introgressions of types I, II, II and Africa 1 into New
World populations have occurred...” I think the authors are suggesting here that Old World lineages
have recombined multiple times with New World strains to produce admixture lines that exist as
chimeras – is that correct?

**AUTHORS' RESPONSE** – This is correct but the admixture pattern (Fig. 3) indicates that these
recombinations were not that frequent.

3. The authors renamed the vast majority of strains to conform with their “domestic” vs. “wild”
designation and geographic origin of the isolate. This was quite a distraction, to sort out which
designation was GT1, Me49, VEG, etc.

**AUTHORS' RESPONSE** – We understand the reviewer's concern and this was a difficult choice to
make. The original designations such as Me49 and VEG are well known for people in the field of
*Toxoplasma*. We renamed the strains to facilitate the understanding for non-specialists and to reach
out to a wider public of readers. Anyway the correspondence with old designations can be made in
Supplementary Table 1

The organization of the Supplemental Tables was difficult to infer. Is it possible to add the common
name to Supplemental Table 2, and also re-sort Supplemental Table 2 to be by multilocus Type, then
by country? Or if the preference is to sort by country, then please sort by multilocus type.

**AUTHORS' RESPONSE** – We modified Supplementary Table 2 according to the reviewer's
recommendation.

4. In the Supplementary File, line 383, the authors state Type III genomes (n=19), but in all other
places within the manuscript, they state that only 18 Type III genomes were analyzed – which is it?

AUTHORS' RESPONSE – We corrected this mistake and the correct number is 19. DcUSA04 (M7741)
corresponds to a strain assumed to belong to type III lineage based on MS markers, and that was
found to be a recombinant strain with WGS (see the “Identifying clonal lineages” section of SI). The
opposite was noticed for DcGabon02: It was a variant of type III on one marker based on MS marker,
but was found to be a true type III strain with WGS.

5. The authors propose a model in which rodents in the New World are multiply co-infected with
different strains of *T. gondii* to support the increased diversity detected in isolates from the New
World. Do the authors have evidence to support such a claim, or have studies been done that they
can reference that indicate such a high frequency of mixed infections to support their proposed
model?

AUTHORS' RESPONSE – The answer to this question can be found in our answer to a question from
reviewer number 2: A host infected with *T. gondii* develops good immunity (not 100% though)
against new infections. However, this immunity appears less efficient when the new infection
involves a highly divergent strain, allowing what is called superinfection to happen. The first study
that has described this phenomenon is a study by Elbez-Rubinstein et al., (2009). Here comes the
importance of defining what a divergent strain is and “divergence” is of course a relative concept.
According to our results and to previous results (Su et al., 2012; Lorenzi et al., 2016), it is clear for
example that wild Amazonian strains are highly divergent from the major domestic types (types I,II,III
and Africa 1). Our results also show that wild type 12 strains (RFLP lineage #5) from North America
are “moderately” divergent from type II strains. Domestic South American strains and wild South
American strains should not be considered one and the same, as the latter are a mixture of wild
South American strains and major domestic lineages. Domestic South American strains have
inherited large portions of their genome from types I, II, III and Africa 1 and this happened very
recently according to our dating estimates. Cross-immunity between these two groups is therefore
likely to occur, but this will likely be determined by the inheritance (or not) of alleles of genes
involved in immunological recognition. Anyway, it is important to recall that superinfected hosts
were rarely found in real life. This rarity fits well with our model, which supports rare recombinations
(see Fig.3), knowing that rare recombinations are probably the result of rare superinfections.

	species scientific name	species name	Geographical area	population estimates	source
Old World	Panthera leo	Lion	Total (Africa)	23 000-39 000	https://dx.doi.org/10.2305/IUCN.UK.2016-3.RLTS.T15951A107265605.en
Old World	Panthera tigris	Tiger	Total (Asia)	2 154-3 159	https://dx.doi.org/10.2305/IUCN.UK.2015-2.RLTS.T15955A50659951.en
Old World	Acinonyx jubatus	Cheetah	Total (Africa)	6 674	https://dx.doi.org/10.2305/IUCN.UK.2015-4.RLTS.T219A50649567.en
Old World	Panthera uncia	Snow Leopard	Total (Asia)	2 710-3 386	https://dx.doi.org/10.2305/IUCN.UK.2017-2.RLTS.T22732A50664030.en
Old World	Panthera pardus orientalis	Amur Leopard	Total (Asia)	<60	https://dx.doi.org/10.2305/IUCN.UK.2020-1.RLTS.T15954A163991139.en
Old World	Panthera pardus nimr	Arabian Leopard	Total (Asia)	45-200	https://dx.doi.org/10.2305/IUCN.UK.2020-1.RLTS.T15954A163991139.en
Old World	Panthera pardus melas	Javan Leopard	Total (Asia)	350-525	https://dx.doi.org/10.2305/IUCN.UK.2020-1.RLTS.T15954A163991139.en
Old World	Panthera pardus kotiya	Sri Lankan Leopard	Total (Asia)	700-950	https://dx.doi.org/10.2305/IUCN.UK.2020-1.RLTS.T15954A163991139.en
Old World	Panthera pardus saxicolor	Persian Leopard	Total (Asia)	800-1 000	https://dx.doi.org/10.2305/IUCN.UK.2020-1.RLTS.T15954A163991139.en
Old World	Lynx pardinus	Iberian Lynx	Total (Europe)	156	https://dx.doi.org/10.2305/IUCN.UK.2015-2.RLTS.T12520A174111773.en
Old World	Lynx lynx	Eurasian Lynx	Europe	9 000-10 000	https://www.iucnredlist.org/species/12519/121707666
Old World	Lynx lynx	Eurasian Lynx	Russia	22 500	https://www.iucnredlist.org/species/12519/121707667
Old World	Lynx lynx	Eurasian Lynx	China	27 000	https://www.iucnredlist.org/species/12519/121707668
Old World	Felis silvestris	Wild cat	Scotland	1 000-4 000	https://www.iucnredlist.org/species/60354712/12918931#population
Old World	Felis silvestris	Wild cat	Germany	1 700-5 000	https://www.iucnredlist.org/species/60354712/12918931#population
Old World	Felis silvestris	Wild cat	Poland	150-200	https://www.iucnredlist.org/species/60354712/12918931#population
Old World	Felis silvestris	Wild cat	Slovakia	1 500	https://www.iucnredlist.org/species/60354712/12918931#population
New World	Lynx rufus	Bobcat	USA	2 352 276-3 571 681	https://dx.doi.org/10.2305/IUCN.UK.2016-1.RLTS.T12521A50655874.en
New World	Leopardus pardalis	Ocelot	Total	1 500 000-3 000 000	https://animaldiversity.org/accounts/Leopardus_pardalis/#9dc6332bb536e3df08677ec82
New World	Panthera onca	Jaguar	Total	64 000	https://dx.doi.org/10.2305/IUCN.UK.2017-3.RLTS.T15953A50658693.en
New World	Puma concolor	Cougar	Total	50 000	https://dx.doi.org/10.2305/IUCN.UK.2015-4.RLTS.T18868A50663436.en

REFERENCES

- Bertranpetit, E., Jombart, T., Paradis, E., Pena, H., Dubey, J., Su, C., Mercier, A., Devillard, S., Ajzenberg, D., 2017. Phylogeography of *Toxoplasma gondii* points to a South American origin. *Infect. Genet. Evol.* 48, 150–155. <https://doi.org/10.1016/j.meegid.2016.12.020>
- Dabritz, H.A., Conrad, P.A., 2010. Cats and *Toxoplasma*: implications for public health. *Zoonoses and public health* 57, 34–52.
- Dalecky, A., Bâ, K., Piry, S., Lippens, C., Diagne, C.A., Kane, M., Sow, A., Diallo, M., Niang, Y., Konečný, A., 2015. Range expansion of the invasive house mouse *Mus musculus domesticus* in Senegal, West Africa: a synthesis of trapping data over three decades, 1983–2014. *Mammal Review* 45, 176–190.
- Dubey, J.P., Rajendran, C., Ferreira, L.R., Kwok, O.C.H., Sinnott, D., Majumdar, D., Su, C., 2010. A new atypical highly mouse virulent *Toxoplasma gondii* genotype isolated from a wild black bear in Alaska. *J Parasitol* 96, 713–716. <https://doi.org/10.1645/GE-2429.1>
- Dubey, J.P., Sundar, N., Hill, D., Velmurugan, G.V., Bandini, L.A., Kwok, O.C.H., Majumdar, D., Su, C., 2008. High prevalence and abundant atypical genotypes of *Toxoplasma gondii* isolated from lambs destined for human consumption in the USA. *International journal for parasitology* 38, 999–1006.
- Elbez-Rubinstein, A., Ajzenberg, D., Dardé, M.-L., Cohen, R., Dumètre, A., Yera, H., Gondon, E., Janaud, J.-C., Thulliez, P., 2009. Congenital toxoplasmosis and reinfection during pregnancy: case report, strain characterization, experimental model of reinfection, and review. *J. Infect. Dis.* 199, 280–285. <https://doi.org/10.1086/595793>
- Farhat, D.C., Swale, C., Dard, C., Cannella, D., Ortet, P., Barakat, M., Sindikubwabo, F., Belmudes, L., De Bock, P.-J., Couté, Y., 2020. A MORC-driven transcriptional switch controls *Toxoplasma* developmental trajectories and sexual commitment. *Nature microbiology* 5, 570–583.
- Fitak, R.R., Rinkevich, S.E., Culver, M., 2018. Genome-wide analysis of SNPs is consistent with no domestic dog ancestry in the endangered Mexican wolf (*Canis lupus baileyi*). *Journal of Heredity* 109, 372–383.
- Henn, B.M., Botigué, L.R., Gravel, S., Wang, W., Brisbin, A., Byrnes, J.K., Fadhlou-Zid, K., Zalloua, P.A., Moreno-Estrada, A., Bertranpetit, J., 2012. Genomic ancestry of North Africans supports back-to-Africa migrations. *PLoS Genet* 8, e1002397.
- Hima, K., Houémenou, G., Badou, S., Garba, M., Dossou, H.-J., Etougbétché, J., Gauthier, P., Artige, E., Fossati-Gaschignard, O., Gagaré, S., 2019. Native and invasive small mammals in urban habitats along the commercial axis connecting Benin and Niger, West Africa. *Diversity* 11, 238.
- Howe, D.K., Sibley, L.D., 1995. *Toxoplasma gondii* comprises three clonal lineages: correlation of parasite genotype with human disease. *J. Infect. Dis.* 172, 1561–1566.
- Hublin, J.-J., Ben-Ncer, A., Bailey, S.E., Freidline, S.E., Neubauer, S., Skinner, M.M., Bergmann, I., Le Cabec, A., Benazzi, S., Harvati, K., 2017. New fossils from Jebel Irhoud, Morocco and the pan-African origin of *Homo sapiens*. *Nature* 546, 289–292.
- Jewell, M.L., Frenkel, J.K., Johnson, K.M., Reed, V., Ruiz, A., 1972. Development of *Toxoplasma* oocysts in neotropical felidae. *The American journal of tropical medicine and hygiene* 21, 512–517.
- Jiang, T., Shwab, E., Martin, R., Gerhold, R., Rosenthal, B., Dubey, J., Su, C., 2018. A partition of *Toxoplasma gondii* genotypes across spatial gradients and among host species, and decreased parasite diversity towards areas of human settlement in North America [WWW Document]. *International journal for parasitology*. <https://doi.org/10.1016/j.ijpara.2018.01.008>
- Khan, A., Ajzenberg, D., Mercier, A., Demar, M., Simon, S., Dardé, M.L., Wang, Q., Verma, S.K., Rosenthal, B.M., Dubey, J.P., Sibley, L.D., 2014a. Geographic separation of domestic and wild strains of *Toxoplasma gondii* in French Guiana correlates with a monomorphic version of chromosome1a. *PLoS Negl Trop Dis* 8, e3182. <https://doi.org/10.1371/journal.pntd.0003182>

- Khan, A., Dubey, J.P., Su, C., Ajioka, J.W., Rosenthal, B.M., Sibley, L.D., 2011a. Genetic analyses of atypical *Toxoplasma gondii* strains reveal a fourth clonal lineage in North America. *Int. J. Parasitol.* 41, 645–655. <https://doi.org/10.1016/j.ijpara.2011.01.005>
- Khan, A., Fux, B., Su, C., Dubey, J.P., Darde, M.L., Ajioka, J.W., Rosenthal, B.M., Sibley, L.D., 2007. Recent transcontinental sweep of *Toxoplasma gondii* driven by a single monomorphic chromosome. *Proc. Natl. Acad. Sci. U.S.A.* 104, 14872–14877. <https://doi.org/10.1073/pnas.0702356104>
- Khan, A., Miller, N., Roos, D.S., Dubey, J.P., Ajzenberg, D., Dardé, M.L., Ajioka, J.W., Rosenthal, B., Sibley, L.D., 2011b. A monomorphic haplotype of chromosome Ia is associated with widespread success in clonal and nonclonal populations of *Toxoplasma gondii*. *MBio* 2, e00228-11.
- Khan, A., Shaik, J.S., Behnke, M., Wang, Q., Dubey, J.P., Lorenzi, H.A., Ajioka, J.W., Rosenthal, B.M., Sibley, L.D., 2014b. NextGen sequencing reveals short double crossovers contribute disproportionately to genetic diversity in *Toxoplasma gondii*. *BMC genomics* 15, 1–15.
- Kim, K., Kwon, T., Dessie, T., Yoo, D., Mwai, O.A., Jang, J., Sung, S., Lee, S., Salim, B., Jung, J., 2020. The mosaic genome of indigenous African cattle as a unique genetic resource for African pastoralism. *Nature Genetics* 52, 1099–1110.
- Lorenzi, H., Khan, A., Behnke, M.S., Namasivayam, S., Swapna, L.S., Hadjithomas, M., Karamycheva, S., Pinney, D., Brunk, B.P., Ajioka, J.W., Ajzenberg, D., Boothroyd, J.C., Boyle, J.P., Dardé, M.L., Diaz-Miranda, M.A., Dubey, J.P., Fritz, H.M., Gennari, S.M., Gregory, B.D., Kim, K., Saeij, J.P.J., Su, C., White, M.W., Zhu, X.-Q., Howe, D.K., Rosenthal, B.M., Grigg, M.E., Parkinson, J., Liu, L., Kissinger, J.C., Roos, D.S., Sibley, L.D., 2016. Local admixture of amplified and diversified secreted pathogenesis determinants shapes mosaic *Toxoplasma gondii* genomes. *Nat Commun* 7, 10147. <https://doi.org/10.1038/ncomms10147>
- Mercier, A., Ajzenberg, D., Devillard, S., Demar, M.P., de Thoisy, B., Bonnabau, H., Collinet, F., Boukhari, R., Blanchet, D., Simon, S., Carme, B., Dardé, M.-L., 2011. Human impact on genetic diversity of *Toxoplasma gondii*: example of the anthropized environment from French Guiana. *Infect. Genet. Evol.* 11, 1378–1387. <https://doi.org/10.1016/j.meegid.2011.05.003>
- Mercier, A., Devillard, S., Ngoubangoye, B., Bonnabau, H., Bañuls, A.-L., Durand, P., Salle, B., Ajzenberg, D., Dardé, M.-L., 2010. Additional haplogroups of *Toxoplasma gondii* out of Africa: population structure and mouse-virulence of strains from Gabon. *PLoS Negl Trop Dis* 4, e876. <https://doi.org/10.1371/journal.pntd.0000876>
- Miller, N.L., Frenkel, J.K., Dubey, J.P., 1972. Oral infections with *Toxoplasma* cysts and oocysts in felines, other mammals, and in birds. *The Journal of parasitology* 928–937.
- Minot, S., Melo, M.B., Li, F., Lu, D., Niedelman, W., Levine, S.S., Saeij, J.P., 2012. Admixture and recombination among *Toxoplasma gondii* lineages explain global genome diversity. *Proceedings of the National Academy of Sciences* 109, 13458–13463.
- Pena, H.F.J., Gennari, S.M., Dubey, J.P., Su, C., 2008. Population structure and mouse-virulence of *Toxoplasma gondii* in Brazil. *Int. J. Parasitol.* 38, 561–569. <https://doi.org/10.1016/j.ijpara.2007.09.004>
- Ramakrishnan, C., Maier, S., Walker, R.A., Rehauer, H., Joekel, D.E., Winiger, R.R., Basso, W.U., Grigg, M.E., Hehl, A.B., Deplazes, P., 2019. An experimental genetically attenuated live vaccine to prevent transmission of *Toxoplasma gondii* by cats. *Scientific reports* 9, 1–14.
- Shapiro, K., VanWormer, E., Packham, A., Dodd, E., Conrad, P.A., Miller, M., 2019. Type X strains of *Toxoplasma gondii* are virulent for southern sea otters (*Enhydra lutris nereis*) and present in felids from nearby watersheds. *Proceedings of the Royal Society B: Biological Sciences* 286. <https://doi.org/10.1098/rspb.2019.1334>
- Shwab, E.K., Saraf, P., Zhu, X.-Q., Zhou, D.-H., McFerrin, B.M., Ajzenberg, D., Schares, G., Hammond-Aryee, K., van Helden, P., Higgins, S.A., Gerhold, R.W., Rosenthal, B.M., Zhao, X., Dubey, J.P., Su, C., 2018. Human impact on the diversity and virulence of the ubiquitous zoonotic parasite *Toxoplasma gondii*. *Proc Natl Acad Sci U S A* 115, E6956–E6963. <https://doi.org/10.1073/pnas.1722202115>

- Su, C., Khan, A., Zhou, P., Majumdar, D., Ajzenberg, D., Dardé, M.-L., Zhu, X.-Q., Ajioka, J.W., Rosenthal, B.M., Dubey, J.P., Sibley, L.D., 2012. Globally diverse *Toxoplasma gondii* isolates comprise six major clades originating from a small number of distinct ancestral lineages. *Proc. Natl. Acad. Sci. U.S.A.* 109, 5844–5849. <https://doi.org/10.1073/pnas.1203190109>
- VanWormer, E., Conrad, P.A., Miller, M.A., Melli, A.C., Carpenter, T.E., Mazet, J.A., 2013a. *Toxoplasma gondii*, source to sea: higher contribution of domestic felids to terrestrial parasite loading despite lower infection prevalence. *EcoHealth* 10, 277–289.
- VanWormer, E., Conrad, P.A., Miller, M.A., Melli, A.C., Carpenter, T.E., Mazet, J.A., 2013b. *Toxoplasma gondii*, source to sea: higher contribution of domestic felids to terrestrial parasite loading despite lower infection prevalence. *EcoHealth* 10, 277–289.

REVIEWER COMMENTS

Reviewer #1 (Remarks to the Author):

I am satisfied with the authors' responses to my comments and those of the other reviewers.

Reviewer #2 (Remarks to the Author):

The authors have addressed the majority of my comments in their response letter. I would note that given that many of these comments were based on a series of misunderstandings of how the data were being interpreted and what the models being tested really were. Perhaps I'm just a bit slow, but in re-reading the manuscript, I see that the information is there, but I also see how I was confused. I note that the issues haven't been resolved.

What I think threw me for a loop was the word "domestic" and its use both for "domesticated" cats (most of which are feral, and essentially genetically identical to the progenitor "wild" cat population) and for strains that are "cosmopolitan" (? though the fact that the authors used both words early in the manuscript led me to believe they consider the two distinct). So I wasn't exactly sure what the definitions were, and somehow made assumptions that the authors were making a claim as to the origins of *Toxoplasma*, in general. I think this was driven mostly by the authors' focus on the "domestication" of cats as a driving force in the evolution of *Toxoplasma*.

To ensure that other readers don't have the same sort of confusion, perhaps the authors could make it a bit more clear (using very plain, non-scientific-jargon language) what their overarching models are. Please define all of your terms (domestic, cosmopolitan) to ensure that it's clear to your reader what your study's goals and conclusions are. Assume everyone is as slow and easily confused as I am.

I'm still not sure I buy the arguments made by the authors w/r/t domesticated vs. wild cats. When they are talking about felids, they are essentially referring to a handful of species, that we (authors and I) all agree are essentially genetically identical. So the origin of the selective pressure for the "domestication" seems somewhat diffuse. Importantly, other species of wildcats that most certainly don't hybridize with "domesticated" cats (feral or otherwise) are neglected (e.g. bobcats, jaguars, etc.). While these have a much lower number in the wild (esp. in modern Europe) than domesticated cats, they produce orders of magnitude higher cyst loads in their scat (see work by Conrad), so it is difficult to rule out their environmental/evolutionary impact simply based on number.

As above, a bit more spoon-feeding the reader through the reasoning and explicit discussion of alternative models that were discarded would help.

Figures are a bit better, but all figures still have tiny text, so are largely illegible. Perhaps the authors could ensure that the printed version of all figures have the smallest font sized at 9 pt. If text cannot be fit into the regions as labels, symbols can be used and explained either in an inset legend (better) or in the figure legends themselves (ok, but not preferable).

Reviewer #1 (Remarks to the Author):

I am satisfied with the authors' responses to my comments and those of the other reviewers.

AUTHORS' RESPONSE – We are grateful to the reviewer for accepting our work.

Reviewer #2 (Remarks to the Author):

The authors have addressed the majority of my comments in their response letter. I would note that given that many of these comments were based on a series of misunderstandings of how the data were being interpreted and what the models being tested really were. Perhaps I'm just a bit slow, but in re-reading the manuscript, I see that the information is there, but I also see how I was confused. I note that the issues haven't been resolved.

AUTHORS' RESPONSE – We are sincerely grateful to the reviewer for accepting to revise our work. His/her comments really contributed to the improvement of our manuscript and brought our attention to certain parts that were lacking clarity.

What I think threw me for a loop was the word "domestic" and its use both for "domesticated" cats (most of which are feral, and essentially genetically identical to the progenitor "wild" cat population) and for strains that are "cosmopolitan" (? though the fact that the authors used both words early in the manuscript led me to believe they consider the two distinct). So I wasn't exactly sure what the definitions were, and somehow made assumptions that the authors were making a claim as to the origins of *Toxoplasma*, in general. I think this was driven mostly by the authors' focus on the "domestication" of cats as a driving force in the evolution of *Toxoplasma*. To ensure that other readers don't have the same sort of confusion, perhaps the authors could make it a bit more clear (using very plain, non-scientific-jargon language) what their overarching models are. Please define all of your terms (domestic, cosmopolitan) to ensure that it's clear to your reader what your study's goals and conclusions are. Assume everyone is as slow and easily confused as I am.

AUTHORS' RESPONSE – We thank the reviewer for this important comment. We acknowledge that the way we presented the results at first was confusing. The word "cosmopolitan" was now replaced by "intercontinental" for simplification. This term is defined in the introduction of the new version of the manuscript. More importantly, we now chose to get rid of our *a priori* classification of strains according to ecotype (domestic or wild). From the RESULTS section onwards (*no a priori* classification), only *T. gondii* populations restricted to the wild environment and found genetically distinct were designated as wild (e.g. Amazonian and Pan-American populations) for more convenience in writing when referring to them. We think that this will improve the clarity of our manuscript and ensure that readers do not get the impression that we classify strains to fit to our model.

We agree with the reviewer about the high degree of genetic identity between the domestic cat *Felis catus* (either being pet, stray or feral cats) and the wildcat *Felis silvestris*. We argue that *T. gondii*

"domestication" was essentially driven by the occupation of the new niches offered by cat domestication and proliferation, and the collapse of original (wild) niches due to the decline of wildcats (*Felis silvestris*) populations. For the host, we observed an "environmental shift" rather than a "genetic shift" during its domestication process, and the parasite "accompanied" its host in its new environment without a need for acquisition of novel adaptations.

I'm still not sure I buy the arguments made by the authors w/r/t domesticated vs. wild cats. When they are talking about felids, they are essentially referring to a handful of species, that we (authors and I) all agree are essentially genetically identical. So the origin of the selective pressure for the "domestication" seems somewhat diffuse.

AUTHORS' RESPONSE – Although felids constitute a family of related species, the Most Recent Common Ancestor (MRCA) of the current 38 felid species described worldwide (Sunquist and Sunquist, 2017) is estimated to have lived in Asia about 11 million years ago (Mya) (O'Brien and Johnson, 2007). This is a sufficiently long time for important evolutions to occur.

In terms of genomic similarity, genetic distances between members of Felidae family (0.00102 on average) are comparable to those of Hominidae (0.00141 on average and includes the great apes — that is, the orangutans, the gorillas, the chimpanzees and bonobos (Pan)— as well as human beings) and Bovidae (0.00133 on average and includes cattle, bison, buffalo, antelopes, and goat-antelopes) (Kim et al., 2016).

Studies reporting evidence of differential adaptations of felid species to *T. gondii* are very scarce and we have cited the experimental studies reporting these elements in our manuscript (Jewell et al., 1972; Miller et al., 1972; Khan et al., 2014). A number of other studies report very interesting results indicating important evolutions of certain felids species toward *T. gondii*, leading to specific adaptations. For example, felid species found in extreme environments such as Pallas cats (*Otocolobus manul*) and sand cats (*Felis margarita*) often die from *T. gondii* infection when exposed to the parasite in places such as zoos for example (Basso et al., 2005; Dubey, 2022; Dubey et al., 2010). This is an extremely rare event in other species of felids. For example, sand cats were found to be killed by strains of Africa 4 lineage (RFLP lineage #20) (Dubey et al., 2010), a non-pathogenic lineage in domestic cats, mice and humans. Although *T. gondii* is considered as a ubiquitous parasite, oocysts need a minimum degree of heat and humidity to sporulate (Lélu et al., 2012) and become infectious. These conditions are not present in extreme environments such as high altitudes and deserts where above mentioned felid species are found, making a cycle of *T. gondii* transmission in these environments unlikely to take place. It is therefore likely that these felid species, not being exposed to *T. gondii* for long times, have lost their adaptation to this parasite.

Studies reporting evidence of differential adaptations of felid species to other pathogens are also available. For example, the feline immunodeficiency virus (FIV) is highly pathogenic in domestic cats whereas it occurs with no or little pathogenesis in other felid species (O'Brien et al., 2006).

Importantly, other species of wildcats that most certainly don't hybridize with "domesticated" cats (feral or otherwise) are neglected (e.g. bobcats, jaguars, etc.).

AUTHORS' RESPONSE – We rather argue that these species are probably efficiently transmitting *T. gondii* populations found in their environment, which often carry divergent haplotypes at the ~100 kb genomic region of chromosome 1a (also refer to Supplementary Discussion 2).

While these have a much lower number in the wild (esp. in modern Europe) than domesticated cats, they produce orders of magnitude higher cyst loads in their scat (see work by Conrad), so it is difficult to rule out their environmental/evolutionary impact simply based on number.

AUTHORS' RESPONSE – Too few reports are available from wild felids to carry robust comparison of oocyst number between species. These reports recorded substantial variability in oocyst concentration in faeces of wild felids ranging from dozens of thousands to millions per gram of faeces (Aramini et al., 1998; Dorny and Franssen, 1989). Overall, these concentrations are comparable to those observed in domestic cats (Dubey et al., 1970). In certain cases, oocyst concentrations in faeces of wild felids can be very low and go undetected by microscopy, although this may be also explained by parasitic strain factors (Bolais et al., 2022).

In the specific context studied by Conrad and colleagues, we found no argument supporting higher oocyst concentrations in wild felids relative to domestic cats. Oocyst shedding frequency in cats (managed and unmanaged feral cats) was estimated to be lower than in bobcats but higher than in mountain lions, indicating no domestic-versus-wild pattern. We quote the discussion of one of their studies (VanWormer et al., 2013): “Diet likely also influences oocyst shedding, but whether or not an animal consumes a prey-based diet may not be completely sufficient for explaining shedding dynamics. The highest prevalences of *T. gondii*-like and confirmed *T. gondii* oocyst shedding were found in unmanaged feral cats and bobcats, both of which subsist predominantly on wild prey. The odds of shedding were significantly higher in these groups than in managed feral cats, whose oocyst shedding estimates were consistent with prior reports (Jones and Dubey 2010). Lower oocyst shedding in managed feral cats is not surprising, given their likely lower consumption of potential intermediate hosts. However, the low levels of oocyst shedding by mountain lions, which consume predominantly wild prey, are more challenging to interpret. Unlike bobcats and unmanaged feral cats, the odds of oocyst shedding were not significantly different in mountain lions compared to feral cats. (...) Unmanaged feral cats, bobcats, and mountain lions in our study had higher prevalences of *T. gondii* infection and shedding than managed feral cats. However, their smaller population sizes likely limit their relative contribution to coastal California oocyst load. The total number of outdoor pet cats and managed feral cats was estimated to be over 75 times larger than the number of wild felids in the 5-kmwide coastal terrestrial environment bordering the sea otter range (Table 3). The estimated numbers of infected and shedding outdoor pet cats and managed feral cats were also drastically higher than those of unmanaged feral cats, mountain lions, and bobcats.” We note that unmanaged feral cats are essentially domestic cats (*Felis catus*) that were not socialized to humans.

As above, a bit more spoon-feeding the reader through the reasoning and explicit discussion of alternative models that were discarded would help.

AUTHORS' RESPONSE – We dedicated a full section (Supplementary Discussion 3) to discuss a number of alternative scenarios.

Figures are a bit better, but all figures still have tiny text, so are largely illegible. Perhaps the authors could ensure that the printed version of all figures have the smallest font sized at 9 pt. If text cannot

be fit into the regions as labels, symbols can be used and explained either in an inset legend (better) or in the figure legends themselves (ok, but not preferable).

AUTHORS' RESPONSE – We modified the figures according to the reviewer's recommendation.

AUTHORS' ADDITIONAL NOTE– We carried out methodological modifications of the TMRCA analysis in response to the editor's request for more clarity in presenting the methodology used in the part. In this new version of the manuscript, we chose to use BEAST software instead of Tang's equation for divergence time estimations. BEAST (Drummond & Rambaut (2007) *BMC evolutionary biology*) is a well-recognized and proven tool for dating purposes (Google Scholar citations = 12,681). It is widely used in high quality journals. It generates time-scaled phylogenetic trees that are much more straightforward to understand compared to the presentation initially provided for the results of Tang's equation. No modifications were made in our estimates of *T. gondii* mutation rate and generation time, which were used to calibrate BEAST. In consequence, the estimates obtained using this tool were essentially similar to those initially obtained with Tang's equation, with the advantage of providing mean values and confidence intervals for our estimates. Our interpretations and conclusions remain the same. For more clarity, we added a figurative illustration of potential biases or artefacts associated to the dating methodology used in this study (Supplementary Fig. 8).

REFERENCES

- Aramini, J., Stephen, C., Dubey, J., 1998. *Toxoplasma gondii* in Vancouver Island cougars (*Felis concolor vancouverensis*): serology and oocyst shedding [WWW Document]. The Journal of parasitology. URL <https://pubmed.ncbi.nlm.nih.gov/9576522/> (accessed 9.23.20).
- Basso, W., Edelhofer, R., Zenker, W., Möstl, K., Kübber-Heiss, A., Prosl, H., 2005. Toxoplasmosis in Pallas' cats (*Otocolobus manul*) raised in captivity. *Parasitology* 130, 293–299.
- Bolais, P.F., Galal, L., Cronemberger, C., Pereira, F. de A., Barbosa, A. da S., Dib, L.V., Amendoeira, M.R.R., Dardé, M.-L., Mercier, A., 2022. *Toxoplasma gondii* in the faeces of wild felids from the Atlantic forest, Brazil.
- Dorny, P., Fransen, J., 1989. Toxoplasmosis in a Siberian tiger (*Panthera tigris altaica*). *Veterinary Record (United Kingdom)*.
- Dubey, J.P., 2022. Clinical toxoplasmosis in zoo animals and its management. *Emerging Animal Species* 100002.
- Dubey, J.P., Miller, N.L., Frenkel, J.K., 1970. The *Toxoplasma gondii* oocyst from cat feces. *Journal of Experimental Medicine* 132, 636–662.
- Dubey, J.P., Pas, A., Rajendran, C., Kwok, O.C.H., Ferreira, L.R., Martins, J., Hebel, C., Hammer, S., Su, C., 2010. Toxoplasmosis in Sand cats (*Felis margarita*) and other animals in the Breeding Centre for Endangered Arabian Wildlife in the United Arab Emirates and Al Wabra Wildlife Preservation, the State of Qatar. *Vet. Parasitol.* 172, 195–203. <https://doi.org/10.1016/j.vetpar.2010.05.013>
- Jewell, M.L., Frenkel, J.K., Johnson, K.M., Reed, V., Ruiz, A., 1972. Development of *Toxoplasma* oocysts in neotropical felidae. *The American journal of tropical medicine and hygiene* 21, 512–517.
- Khan, A., Ajzenberg, D., Mercier, A., Demar, M., Simon, S., Dardé, M.L., Wang, Q., Verma, S.K., Rosenthal, B.M., Dubey, J.P., Sibley, L.D., 2014. Geographic separation of domestic and wild strains of *Toxoplasma gondii* in French Guiana correlates with a monomorphic version of chromosome1a. *PLoS Negl Trop Dis* 8, e3182. <https://doi.org/10.1371/journal.pntd.0003182>

- Kim, S., Cho, Y.S., Kim, H.-M., Chung, O., Kim, H., Jho, S., Seomun, H., Kim, J., Bang, W.Y., Kim, C., 2016. Comparison of carnivore, omnivore, and herbivore mammalian genomes with a new leopard assembly. *Genome biology* 17, 1–12.
- Lélu, M., Villena, I., Dardé, M.-L., Aubert, D., Geers, R., Dupuis, E., Marnef, F., Poulle, M.-L., Gotteland, C., Dumètre, A., 2012. Quantitative estimation of the viability of *Toxoplasma gondii* oocysts in soil. *Applied and environmental microbiology* 78, 5127–5132.
- Miller, N.L., Frenkel, J.K., Dubey, J.P., 1972. Oral infections with *Toxoplasma* cysts and oocysts in felines, other mammals, and in birds. *The Journal of parasitology* 928–937.
- O’Brien, S.J., Johnson, W.E., 2007. The evolution of cats. *Scientific American* 297, 68–75.
- O’Brien, S.J., Troyer, J.L., Roelke, M., Marker, L., Pecon-Slattery, J., 2006. Plagues and adaptation: Lessons from the Felidae models for SARS and AIDS. *Biological conservation* 131, 255–267.
- Sunquist, M., Sunquist, F., 2017. *Wild cats of the world*. University of Chicago press.
- VanWormer, E., Conrad, P.A., Miller, M.A., Melli, A.C., Carpenter, T.E., Mazet, J.A., 2013. *Toxoplasma gondii*, source to sea: higher contribution of domestic felids to terrestrial parasite loading despite lower infection prevalence. *EcoHealth* 10, 277–289.

REVIEWERS' COMMENTS

Reviewer #2 (Remarks to the Author):

The reviewers have addressed my concerns in their edits and reformats of the manuscript.